electrical engineering/structural engineering

dynamic stiffness, series of spans, transmission line, mode, natural frequency

**Author for correspondence:**
Xiaohui Liu
e-mail: cqdxlxh@126.com

# Free vibration analysis of transmission lines based on the dynamic stiffness method

Xiaohui Liu[1], You Hu[1] and Mengqi Cai[2]

[1]College of Civil Engineering, Chongqing Jiaotong University, Chongqing 400074, People's Republic of China
[2]College of Architecture and Civil Engineering, Chengdu University, Chengdu 610106, People's Republic of China

XL, 0000-0002-2299-9279; MC, 0000-0002-1941-0179

An improved mathematical model used to study the coupling characteristic of the multi-span transmission lines is developed. Based on the solution method for single-span cable, an expression for the dynamic stiffness of two-span transmission lines with an arbitrary inclination angle is formulated. The continuity of displacements and forces at a suspension point is used to derive the dynamic stiffness. Interactions between insulator strings and adjacent spans are accommodated. Considering the infinite dynamic stiffness corresponds to the natural frequencies of the transmission lines, the finite-element method (FEM) is employed to assess the validity of the dynamic stiffness. In the numerical investigation, attention is focused on the effect of the inclination angles, Irvine parameter, insulator string length and damping parameter. In addition, the modal function corresponding to the natural frequencies is derived. Then, the results of comprehensive parametric studies are presented and discussed. Special attention is paid to the effect of the Irvine parameter and damping parameter on the in-plane modal shapes. Finally, according to the theoretical model of two-span transmission lines, the generalized dynamic stiffness of transmission lines with an arbitrary number of spans and inclination angles is derived. The method can be used as the basis of the vibration analysis on a wide variety of multi-span transmission lines.

## 1. Introduction

Dynamic stiffness can be applied to evaluate the dynamic response of cables and of cable-supported structures in any cable-structure system. Galloping is associated with the high amplitude and additional tension in the cable, which can lead to damage to the electrical power network, such as short circuiting, failure of sub-conductors and spacers, and even the collapse of towers [1,2]. Figure 1 shows the damage to the spacer and the tower arm due

**Figure 1.** Damaged overhead transmission lines. (*a*) Tower arm damage. (*b*) Spacer damage.

to transmission line galloping. In January 2018, with the influence of cold temperatures and snow in central and eastern China, some mechanical damage to transmission lines caused by galloping occurred. Therefore, anti-galloping is fundamental in maintaining a reliable service of overhead transmission lines.

Since some galloping characteristics are similar to those of free vibrating suspended cable, the fundamental and most important aspect is to study the dynamic response of free vibrating suspended cable. For instance, the frequency and modal shape of a galloping conductor are similar to those of free vibrating suspended cable, and internal resonance phenomena of suspended cable induced by galloping can be observed in free vibrating suspended cable. Initially, Irvine [3] presented a linear theory for free vibration of a horizontal single-span cable fixed at both ends and discussed the effect of the geometrical-elastic parameter $\lambda$ on the modal shape and natural frequency. Then, ignoring the weight component parallel to the chord of an inclined cable, Irvine [4] proposed a linear theory for the free vibration of inclination cable fixed at both ends. Yamaguchi [5] analysed the effects of the weight component parallel to the chord of an inclined cable on the natural frequencies and modal shapes. In addition, the effects of the weight component from other perspectives have been studied by many scholars [6–8]. It is known that the additional tension induced by galloping may cause damage to towers. To solve this problem, the dynamic stiffness of single-span cable has been presented to evaluate the resistance to deformation of the single-span cable [9–12]. The dynamic stiffness method is often considered since the dynamic stiffness can be used to identify natural frequencies of a cable system and evaluate the effect of different modes on additional tension [13–15]. In recent years, the dynamic stiffness method was applied to various types of beams and plates. Banerjee *et al.* [16,17] investigated the free bending vibration of rotating tapered beams and the free vibration of three-layered symmetric sandwich beams using the dynamic stiffness method. Later, this dynamic stiffness was employed by Banerjee to study the free vibration characteristics of an axial-bending coupled beam [18]. Li *et al.* [19] studied the effect of axial compressive force on natural frequencies of laminated composite beam using the dynamic stiffness method. Li *et al.* [20] used the dynamic stiffness method to perform the exact vibration and buckling analysis of laminated composite beams. Bozyigit *et al.* [21] studied the free vibration analysis of axially moving beams. In this study, the first three natural frequencies of beam calculated by using the dynamic stiffness method and DSM are obtained and some of the results are compared with ABAQUS. Kolarevic *et al.* [22] used the dynamic stiffness method to obtain natural frequencies of individual plates and plate assemblies with arbitrary boundary conditions. Xu *et al.* [23] analysed the influence of bending cracks and steel-concrete bond damage on the dynamic stiffness. Li *et al.* [24] proposed the dynamic stiffness method to analyse the free vibration of multi-span pipe conveying fluid.

Nonlinear vibrations of single-span cable appear in various engineering problems. These studies on free vibrations of single-span cable allow one to understand the response to nonlinear vibration. For instance, two or more linear natural frequencies that are nearly commensurable at certain values of the geometrical parameter, internal resonance and multimodal vibration appear [25,26]. By means of the natural frequencies and modes, a multimode discretization was presented for the investigation of the nonlinear response of continuous systems [27,28]. Since then, internal resonances and bifurcations of single-span cable using the multimode discretization have been studied by many scholars [29,30]. In fact, internal resonance may arise in galloping of transmission lines. The nonlinear galloping characteristics and corresponding instability of iced suspended cables have been investigated by using a combination of the methods of multiple scales and Galerkin, when the internal resonances

exist [31,32]. The numerical results of galloping obtained by using the nonlinear finite-element method (FEM) further support this contention that internal resonances exist in galloping of a single-span cable [33].

A number of investigations on internal resonances of galloping have been published [34–37]; however, these methods have been restricted to single span with fixed ends. In addition, Zhou et al. [38] studied rain–wind-induced galloping phenomena on the single-span transmission line. Actually, the multi-span section, which consists of a series of coupled spans, is the most representative structure in overhead transmission lines. In a multi-span section, a dynamic coupling between adjacent spans may occur because of the influence of the swing of the insulator string at the tangent towers [39,40]. Initially, in order to simplify the calculation, a single-span model with spring boundary conditions was presented, and this stiffness of the springs representing the adjacent spans and insulator stings was obtained by using the dynamic stiffness method [41]. Wang et al. [42] pointed out that the galloping amplitude on single-span mode with spring boundary conditions is larger than that with fixed boundary conditions. However, the equivalent spring does not reflect the inertia of the adjacent spans. Recently, Xie et al. [43] studied the effects of the span ratio on the natural frequencies in a two-span section, and Yi et al. [44] studied the vibration characteristics of a three-span mass-carrying cable with multiple pulley supports.

Although there are a number of studies that have addressed the free vibrations of horizontal cable, to the knowledge of the authors of this paper, studies on the dynamic stiffness and natural frequencies in multi-span transmission lines with inclination angles have not been reported. In this paper, the dynamic stiffness of two-span transmission lines is studied for the first time, which adopts the subsystem synthesis method and boundary conditions. Then, using the dynamic stiffness, the effects of inclination angle and insulator string length on the natural frequencies are discussed. Special attention is paid to the natural frequencies and modes in transition regions, which are predicted by using the dynamic stiffness. Based on that, the theoretical formula for the dynamic stiffness of arbitrary multi-span transmission lines with inclination angles is proposed.

# 2. Dynamic stiffness and modal function of two-span transmission lines

## 2.1. Theoretical formula for the dynamic stiffness

The two-span transmission lines, which are the simplest multi-span transmission lines, can exhibit coupling characteristics between adjacent spans. Therefore, a model of two-span transmission lines with an insulator string is developed to compute dynamic stiffness, as shown in figure 2. Here, it is assumed that point $C$ is suspended from a dead-end tower. Ignoring the displacement of the dead-end tower under the dynamic tension of a conductor, point $C$ is fixed by using a hinge constraint. Considering point $O$ suspended from a tangent tower, ignoring the displacement of the tangent tower, the displacement at point $O$ is also limited by using a hinge constraint. Since point $A$ may be suspended from an insulator string (such as the $B$ end), in the model point $A$ is limited by using a sliding hinge. Every span in transmission lines is suspended from two supports located at different elevations. $l_1$ and $l_2$ are the lengths of the straight line between two supports in the first and second spans, respectively. $l_{1H}$ and $l_{2H}$ are the horizontal distances between two supports in the first and second spans, respectively. $h_1$ and $h_2$ are height differences between two supports in the first and second spans, respectively. $\beta_1$ and $\beta_2$ are inclination angles of the first and second spans, respectively. In figure 2, the dotted line is a static configuration of two-span transmission lines under self-weight. The static equilibrium profile of every span in transmission lines is usually assumed to be the parabola. The local coordinate system of the first span with the origin taken at the left end of the first span is set up, so that $x_1$ measures distance along the chord from the origin $A$ and $y_1$ measures distance to the profile from the chord and perpendicular to it. The local coordinate system $(Bx_2y_2)$ of the second span is defined similarly. For each span with small sag rations, ignoring the effect of the self-weight component parallel to the chord, the deflections of the first and second spans at their positions of static equilibrium are then given by [4]

$$y_1 = \frac{q\cos\beta_1 l_1^2}{2H_1}\left(\frac{x_1}{l_1} - \left(\frac{x_1}{l_1}\right)^2\right); \quad y_2 = \frac{q\cos\beta_2 l_2^2}{2H_2}\left(\frac{x_2}{l_2} - \left(\frac{x_2}{l_2}\right)^2\right), \tag{2.1}$$

where $q$ is self-weight per unit length, $H_1$ and $H_2$ are axial tension components parallel to the chords of the first and second spans, respectively.

Considering the effect of the adjacent span on the one of interest, point $A$ is subjected to an additional harmonically external force $\Delta F = \Delta \bar{F} e^{i\omega t}$, the dynamic configuration of the two-span transmission lines is plotted in solid line as shown in figure 2. It can be assumed that the corresponding displacement

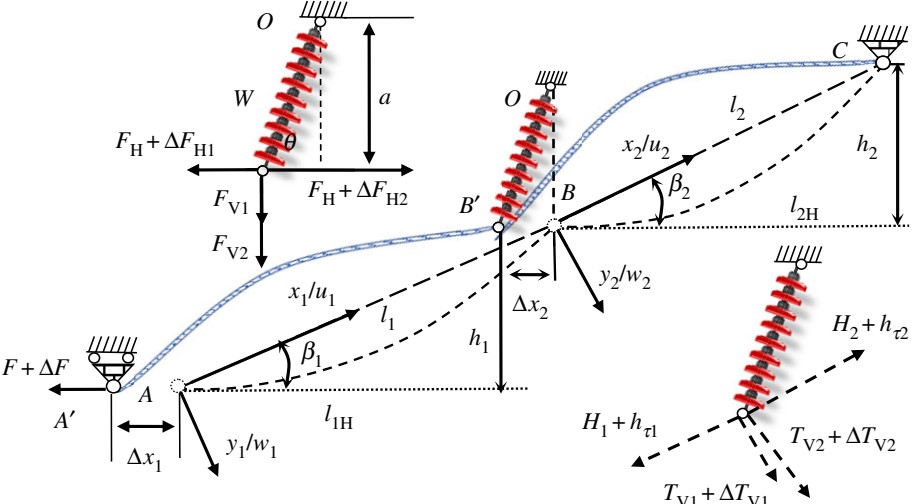

**Figure 2.** A schematic of two-span transmission lines with insulator string under external excitation.

amplitude $\Delta x_1$ is small. Here, all dynamic displacements have been assumed to be small such that the response to the equilibrium position is linear. $w_1$ and $w_2$ represent the normal displacements of the first and second spans, respectively. For two-span transmission lines, the governing equations of in-plane motion with respect to their equilibrium positions can be obtained as [4–6]

$$H_1 \frac{\partial^2 w_1}{\partial x_1^2} + h_{\tau 1} \frac{d^2 y_1}{dx_1^2} = m \frac{\partial^2 w_1}{\partial t^2} + c \frac{\partial w_1}{\partial t} \tag{2.2a}$$

and

$$H_2 \frac{\partial^2 w_2}{\partial x_2^1} + h_{\tau 2} \frac{d^2 y_2}{dx_2^2} = m \frac{\partial^2 w_2}{\partial t^2} + c \frac{\partial w_2}{\partial t}, \tag{2.2b}$$

where $m$ is the mass of the conductor per unit length, $c$ is the coefficient of viscous damping per unit length, $h_{\tau 1}$ and $h_{\tau 2}$ are the dynamic increments in $H_1$ and $H_2$, respectively. In order to simplify the calculation, according to the actual state, the transmission lines are assumed so that the conductor tension is much greater than the self-weight component parallel to the chord of every span, and only the normal components of the inertia and damping forces are of importance. Based on the fundamental assumption, neglecting the self-weight and initial force components parallel to the chord, the tension $H_1$ (or $H_2$) and corresponding dynamic increment $h_{\tau 1}$ (or $h_{\tau 2}$) along the chord are independent of the position coordinate, $x$. Considering the small displacements at both ends of a span and conductor strain, the tension increments of the first and second spans are given by the following equations (2.3a,b) [3]:

$$h_{\tau 1} = \frac{AE}{l_{e1}} \left[ (u_1(l_1) - u_1(0)) - \frac{1}{2} \frac{q_{y1} l_1}{H_1} (w_1(l_1) + w_1(0)) + \frac{q_{y1}}{H_1} \int_0^{l_1} w_1(x)\, dx \right] \tag{2.3a}$$

and

$$h_{\tau 2} = \frac{AE}{l_{e2}} \left[ (u_2(l_2) - u_2(0)) - \frac{1}{2} \frac{q_{y2} l_2}{H_2} (w_2(l_2) + w_2(0)) + \frac{q_{y2}}{H_2} \int_0^{l_2} w_2(x)\, dx \right], \tag{2.3b}$$

where

$$q_{y1} = q \cos \beta_1 = mg \cos \beta_1; \quad q_{y2} = q \cos \beta_2, \tag{2.4}$$

$u_1(0)$, $u_1(l_1)$, $w_1(0)$ and $w_1(l_1)$ represent prescribed motion of left and right ends of the first span, respectively. $l_{e1}$ and $l_{e2}$ are the effective cable lengths of the first and second spans, respectively [9]. $u_2(0)$, $u_2(l_2)$, $w_2(0)$ and $w_2(l_2)$ are similar to those of the second span. The boundary conditions at points A and C are obtained by

$$u_1(0) = \Delta x_1 \cos \beta_1; \; w_1(0) = \Delta x_1 \sin \beta_1, \tag{2.5a}$$

and

$$u_2(l_2) = 0; \; w_2(l_2) = 0 . \tag{2.5b}$$

Taking into account the continuity conditions of displacements, the displacements at point B are obtained by

$$u_1(l_1) = \Delta x_2 \cos\beta_1; \quad u_2(0) = \Delta x_2 \cos\beta_2 \tag{2.6a}$$

$$w_1(l_1) = \Delta x_2 \sin\beta_1; \quad w_2(0) = \Delta x_2 \sin\beta_2. \tag{2.6b}$$

The displacement $w_1$ of the first span consists of two parts, the first part is rigid rotation displacement and the second part represents the displacement induced by conductor deformation. The displacement $w_2$ of the second span is similar to that of the first span. $w_1$ and $w_2$ are given by

$$w_1 = w_1(0) + [w_1(l_1) - w_1(0)]\frac{x}{l_1} + w_{t1}(x,t) \tag{2.7a}$$

and

$$w_2 = w_2(0) + [w_2(l_2) - w_2(0)]\frac{x}{l_2} + w_{t2}(x,t), \tag{2.7b}$$

where $w_{t1}$ and $w_{t2}$ represent the displacements induced by the conductor deformation. Substituting equation (2.7a,b) and equation (2.1) into equation (2.2a,b), we obtain

$$H_1 \frac{\partial^2 w_{t1}}{\partial x_1^2} = m\frac{\partial^2 w_{t1}}{\partial t^2} + c\frac{\partial w_{t1}}{\partial t} + h_{\tau1}\frac{q_{y1}}{H_1} + m\frac{\partial^2 w_{r1}}{\partial t^2} + c\frac{\partial w_{r1}}{\partial t} \tag{2.8a}$$

and

$$H_2 \frac{\partial^2 w_{t2}}{\partial x_2^2} = m\frac{\partial^2 w_{t2}}{\partial t^2} + c\frac{\partial w_{t2}}{\partial t} + h_{\tau2}\frac{q_{y2}}{H_2} + m\frac{\partial^2 w_{r2}}{\partial t^2} + c\frac{\partial w_{r2}}{\partial t}, \tag{2.8b}$$

where

$$w_{r1} = \Delta x_1 \sin\beta_1 + [\Delta x_2 \sin\beta_1 - \Delta x_1 \sin\beta_1]\frac{x}{l_1} \tag{2.9a}$$

and

$$w_{r2} = \Delta x_2 \sin\beta_2 - \Delta x_2 \sin\beta_2 \frac{x}{l_2}. \tag{2.9b}$$

Insulator string is an important part of transmission lines and plays an important role in electrical insulation and mechanical support. Owing to the swing of insulator string, the kinetic energies of any span may be transferred to adjacent spans. So, it is necessary to consider the effect of the insulator string on the dynamic motion of transmission lines. Assuming the swing angle $\theta$ of insulator string to be small, the moment equation of the insulator string around the suspension point $O$ is obtained as

$$a(\Delta F_{H2} - \Delta F_{H1}) = J\ddot{\theta} + (F_{V1} + F_{V2})\Delta x_2 + W\frac{\Delta x_2}{2}, \tag{2.10}$$

where $J$ is insulator string moment of inertia, $W$ and $a$ are insulator string mass and length, respectively. The vertical forces $F_{V1}$ and $F_{V2}$ exerted by the first and second spans at $B$, respectively, are shown in figure 2. $F_{V1}$ and $F_{V2}$ are given by

$$F_{V1} = \frac{q}{2}l_1 + H\frac{h_1}{\sqrt{l_{1H}^2 - h_1^2}}; \quad \Delta T_{V2} = h_{\tau2}\frac{dy_2}{dx} + H_2\frac{\partial w_2}{\partial x_2}. \tag{2.11}$$

Limiting now the discussion to transmission lines with small sags and low-amplitude motions, the component of the tension directed along the axis normal to the chord joining the support points is given by

$$\Delta T_{V1} = h_{\tau1}\frac{dy_1}{dx_1} + H_1\frac{\partial w_1}{\partial x_1}; \quad \Delta T_{V2} = h_{\tau2}\frac{dy_2}{dx} + H_2\frac{\partial w_2}{\partial x_2}. \tag{2.12}$$

The horizontal components of the forces acting at the lower point of insulator string as shown in figure 2 are obtained by

$$\Delta F_{H1} = h_{\tau1}\cos\beta_1 + \Delta T_{V}1\sin\beta_1; \quad \Delta F_{H2} = h_{\tau2}\cos\beta_2 + \Delta T_{V2}\sin\beta_2. \tag{2.13}$$

It is noted that equations (2.12) and (2.13) are linearized formulations.

When *A* point in two-span transmission lines is subjected to a harmonic force, the increment displacements and additional tensions are given by

$$\Delta F(t) = \Delta \bar{F} e^{i\omega t}; \quad h_{\tau 1}(t) = \bar{h}_{\tau 1} e^{i\omega t}; \quad h_{\tau 2}(t) = \bar{h}_{\tau 2} e^{i\omega t}, \tag{2.14a}$$

$$w_{t1}(x,t) = \bar{w}_{t1}(x) e^{i\omega t}; \quad w_{t2}(x,t) = \bar{w}_{t2}(x) e^{i\omega t} \tag{2.14b}$$

and

$$\Delta x_1(t) = \Delta \bar{x}_1 e^{i\omega t}; \quad \Delta x_2(t) = \Delta \bar{x}_2 e^{i\omega t}. \tag{2.14c}$$

Using equation (2.14a–c), equation (2.8a,b) becomes

$$\frac{d^2 \bar{w}_{t1}}{d\zeta^2} + \bar{\omega}_1^2 \bar{w}_{t1} = \frac{q_{y_1} \bar{h}_{\tau 1} l_1^2}{H_1^2} - \bar{\omega}_1^2 \Delta \bar{x}_1 \sin \beta_1 - \bar{\omega}_1^2 (\Delta \bar{x}_2 \sin \beta_1 - \Delta \bar{x}_1 \sin \beta_1) \zeta \tag{2.15a}$$

and

$$\frac{d^2 \bar{w}_{t2}}{d\zeta^2} + \bar{\omega}_2^2 \bar{w}_{t2} = \frac{q_{y_2} \bar{h}_{\tau 2} l_2^2}{H_2^2} - \bar{\omega}_2^2 \Delta \bar{x}_2 \sin \beta_2 + \bar{\omega}_2^2 \Delta \bar{x}_2 \zeta \sin \beta_2, \tag{2.15b}$$

where $0 \le \zeta \le 1$, $\bar{\omega}_1^2 = (m\omega^2 - \omega ci) l_1^2 / H_1$, $\bar{\omega}_2^2 = (m\omega^2 - \omega ci) l_2^2 / H_2$.

The solutions of equation (2.15a,b), with the given boundary conditions of equation (2.5a,b), are obtained by

$$\bar{w}_{t1}(x) = \frac{q_{y_1} \bar{h}_{\tau 1} l_1^2}{\bar{\omega}_1^2 H_1^2} \left( 1 - \sin \frac{\bar{\omega}_1}{l_1} x \tan \frac{\bar{\omega}_1}{2} - \cos \frac{\bar{\omega}_1}{l_1} x \right) - \Delta \bar{x}_1 \sin \beta_1 \left( 1 + \frac{1}{\tan \bar{\omega}_1} \sin \frac{\bar{\omega}_1}{l_1} x - \cos \frac{\bar{\omega}_1}{l_1} x - \frac{x}{l_1} \right)$$

$$- \Delta \bar{x}_2 \sin \beta_1 \left( \frac{x}{l_1} - \frac{1}{\sin \bar{\omega}_1} \sin \frac{\bar{\omega}_1}{l_1} x \right) \tag{2.16a}$$

and

$$\bar{w}_{t2}(x) = \frac{q_{y_2} \bar{h}_{\tau 2} l_2^2}{\bar{\omega}_2^2 H_2^2} \left( 1 - \sin \frac{\bar{\omega}_2}{l_2} x \tan \frac{\bar{\omega}_2}{2} - \cos \frac{\bar{\omega}_2}{l_2} x \right)$$

$$- \Delta \bar{x}_2 \sin \beta_2 \left( 1 + \frac{1}{\tan \bar{\omega}_2} \sin \frac{\bar{\omega}_2}{l_2} x - \cos \frac{\bar{\omega}_2}{l_2} x - \frac{x}{l_2} \right), \tag{2.16b}$$

where equation (2.16a,b) represents the modal shape of two-span transmission lines. However, in equation (2.16a,b), there are some unknown parameters to be solved. Substituting equations (2.5a,b), (2.6a,b), (2.13) and (2.16a,b) into equation (2.3a,b), the tension increments are given by

$$\bar{h}_{\tau 1} = \frac{AE}{l_1} \cos \beta_1 \frac{(\Delta \bar{x}_2 - \Delta \bar{x}_1) - (q_{y1} l_1 / 2H_1) A_1 (\Delta \bar{x}_1 + \Delta \bar{x}_2) \tan \beta_1}{1 - (\lambda_1^2 / \bar{\omega}_1^2) A_1} \tag{2.17a}$$

and

$$\bar{h}_{\tau 2} = -\frac{AE}{l_2} \Delta x_2 \cos \beta_2 \frac{1 + (q_{y2} l_2 / 2H_2) A_2 \tan \beta_2}{1 - (\lambda_2^2 / \bar{\omega}_2^2) A_2}, \tag{2.17b}$$

where

$$\lambda_1^2 = \frac{EA}{H_1} \left( \frac{q_{y_1} l_1}{H_1} \right)^2; \quad \lambda_2^2 = \frac{EA}{H_2} \left( \frac{q_{y_2} l_2}{H_2} \right)^2 \tag{2.18a}$$

$$A_1 = 1 - \frac{\tan(\bar{\omega}_1 / 2)}{\bar{\omega}_1 / 2}; \quad A_2 = 1 - \frac{\tan(\bar{\omega}_2 / 2)}{\bar{\omega}_2 / 2}. \tag{2.18b}$$

Substituting equation (2.14a–c) into equation (2.12), $\Delta \bar{T}_{V1}$ and $\Delta \bar{T}_{V2}$ at both ends are given by

$$\Delta \bar{T}_{V1}(0) = \bar{h}_{\tau 1} \frac{q_{y1} l_1}{2H_1} A_1 + H_1 \sin \beta_1 \left( \frac{\Delta \bar{x}_2}{l_1} \cdot \frac{\bar{\omega}_1}{\sin \bar{\omega}_1} - \frac{\Delta \bar{x}_1}{l_1} \frac{\bar{\omega}_1}{\tan \bar{\omega}_1} \right), \tag{2.19a}$$

$$\Delta \bar{T}_{V1}(l_1) = -\bar{h}_{\tau 1} \frac{q_{y1} l_1}{2H_1} A_1 + H_1 \sin \beta_1 \left( \frac{\Delta \bar{x}_2}{l_1} \frac{\bar{\omega}_1}{\tan \bar{\omega}_1} - \frac{\Delta \bar{x}_1}{l_1} \cdot \frac{\bar{\omega}_1}{\sin \bar{\omega}_1} \right) \tag{2.19b}$$

and

$$\Delta \bar{T}_{V2}(0) = \bar{h}_{\tau 2} \frac{q_{y2} l_2}{2H_2} A_2 - H_2 \sin \beta_2 \left( \frac{\Delta \bar{x}_2}{l_2} \frac{\bar{\omega}_2}{\tan \bar{\omega}_2} \right). \tag{2.19c}$$

The vertical force $\Delta \bar{T}_{V1}(0)$ is exerted by the first span at *A*. The vertical force $\Delta \bar{T}_{V1}(l_1)$ is exerted by the first span at *B*. The vertical force $\Delta \bar{T}_{V2}(0)$ is exerted by the second span at *B*.

Substituting equations (2.19*a*−*c*) and (2.14*a*−*c*) into equation (2.13), the horizontal components of the forces acting at $B$, $\Delta \bar{F}_{\mathrm{H1}}$ and $\Delta \bar{F}_{\mathrm{H2}}$, are given by

$$\Delta \bar{F}_{\mathrm{H1}} = B_{22}^1 \Delta \bar{x}_2 - B_{21}^1 \Delta \bar{x}_1; \quad \Delta \bar{F}_{\mathrm{H2}} = -B_{11}^2 \Delta \bar{x}_2, \tag{2.20}$$

where

$$B_{11}^1 = \frac{(AE/l_1)(\cos \beta_1 - (q_{y1} l_1 / 2H_1) A_1 \sin \beta_1)^2}{1 - (\lambda_1^2 / \bar{\omega}_1^2) A_1} + \frac{H_1 \sin^2 \beta_1 \bar{\omega}_1}{l_1 \tan \bar{\omega}_1}, \tag{2.21a}$$

$$B_{21}^1 = \frac{(AE/l_1)(\cos^2 \beta_1 - ((q_{y1} l_1 / 2H_1) A_1 \sin \beta_1)^2)}{1 - (\lambda_1^2 / \bar{\omega}_1^2) A_1} + \frac{H_1 \sin^2 \beta_1 \bar{\omega}_1}{l_1 \sin \bar{\omega}_1}, \tag{2.21b}$$

$$B_{22}^1 = \frac{(AE/l_1)(\cos \beta_1 + (q_{y1} l_1 / 2H_1) A_1 \sin \beta_1)^2}{1 - (\lambda_1^2 / \bar{\omega}_1^2) A_1} + \frac{H_1 \sin^2 \beta_1 \bar{\omega}_1}{l_1 \tan \bar{\omega}_1} \tag{2.21c}$$

and

$$B_{11}^2 = \frac{(AE/l_2)(\cos \beta_2 - (q_{y2} l_2 / 2H_2) A_2 \sin \beta_2)^2}{1 - (\lambda_2^2 / \bar{\omega}_2^2) A_2} + \frac{H_2 \sin^2 \beta_2 \bar{\omega}_2}{l_2 \tan \bar{\omega}_2}. \tag{2.21d}$$

Substituting equations (2.20) and (2.14*a*−*c*) into equation (2.10), combining the equilibrium equation at point $A$, we obtain

$$\Delta \bar{F} = B_{12}^1 \Delta \bar{x}_2 - B_{11}^1 \Delta \bar{x}_1; \quad B^2 \Delta \bar{x}_2 - B_{21}^1 \Delta \bar{x}_1 = 0, \tag{2.22}$$

where $B_{12}^1 = B_{21}^1$, $B_2$ is

$$B_2 = B_{22}^1 + B_{11}^2 + \frac{1}{a}\left[-\frac{\omega^2 J}{a} + (F_{\mathrm{V1}} + F_{\mathrm{V2}}) + \frac{W}{2}\right]. \tag{2.23}$$

The dynamic stiffness, $K_{\mathrm{d}}$, is the ratio of the applied force to the corresponding displacement. From equations (2.22) and (2.23), the dynamic stiffness function could be given by the expression

$$K_{\mathrm{d}} = \mathop{Lim}_{\Delta \bar{x}_1 \to 0} \frac{\Delta \bar{F}}{\Delta \bar{x}_1} = -B_{11}^1 + \frac{B_{12}^1 B_{21}^1}{B_2}. \tag{2.24}$$

The dynamic stiffness of two-span transmission lines, $K_{\mathrm{d}}$, depends on geometrical parameters, materials and excitation frequency. Upon carrying out the limiting transition $\omega \to 0$, equation (2.21*a*−*d*) reduces to

$$B_{21}^1 = B_{12}^1 = B_{22}^1 = \frac{(AE/l_1)(\cos \beta_1)^2}{1 + (\lambda_1^2 / 12)} + \frac{H_1 \sin^2 \beta_1}{l_1}; \quad B_{11}^2 = \frac{(AE/l_2)(\cos \beta_2)^2}{1 + (\lambda_2^2 / 12 \bar{\omega}_2^2)} + \frac{H_2 \sin^2 \beta_2}{l_2}. \tag{2.25}$$

Substituting equation (2.25) into equation (2.24), the equivalent dynamic stiffness $K_{\mathrm{d}}$ can be obtained. The dimensionless frequency $\bar{\omega}_1 / \pi$ corresponding to the peak is the natural frequency of two-span transmission lines, when a peak appears on the dynamic stiffness curve varying with excited frequency $\bar{\omega}_1 / \pi$. Finally, the accuracy of the dynamic stiffness has been verified by comparing with the natural frequency of transmission lines obtained by mean of ABAQUS (FEM software).

## 2.2. Finite-element modelling of two-span transmission lines

In order to verify the validity of the dynamic stiffness function, an FEM model of two-span transmission lines is established. As the conductor diameter is much smaller than the span length, the bending stiffness could be neglected. Thus, the conductor in two-span transmission lines is modelled with truss elements. It is common sense that the exact configuration of a span conductor under self-weight can be depicted with catenary. It is noted that the dynamic stiffness is based on a parabola configuration in order to simplify the calculations. In figure 5, the catenary configuration of the first span under self-weight can be written as

$$y_1 = \frac{H}{p} \mathrm{ch}\left(\frac{p(x_1 - a_1)}{H}\right) - \frac{H}{p} \mathrm{ch}\left(-\frac{p a_1}{H}\right), \tag{2.26}$$

where

$$a_1 = \frac{l_{1\mathrm{H}}}{2} - \frac{H}{p} \mathrm{arc\ sh} \frac{p h_1}{2H \mathrm{sh}(p h_1 / 2H)}, \tag{2.27}$$

where the horizontal component of the conductor tension, $H$, is a constant, and $l_{1\mathrm{H}}$ is the horizontal span length of the first span.

If the first span conductor with initial length $L_{10}$ extends to $L_{10}+\Delta L_1$ under axial tension $T_1$ and the deformation of the conductor is in elastic scope, the initial length of the first span conductor can be expressed as

$$L_{10} = L_1 - \Delta L_1 = L_1 - \int_0^{l_1} \frac{T_1(x_1)}{EA}\,\mathrm{d}x. \tag{2.28}$$

According to equation (2.26), the length of the first span conductor $L_{10}$ and axial tension $T_1$ are given by

$$T_1 = H\mathrm{ch}\frac{p(x_1 - a_1)}{H}; \quad L_1 = \frac{H}{p}\left[\mathrm{sh}\left(\frac{P(l - a_1)}{H}\right) + \mathrm{sh}\left(\frac{pa_1}{H}\right)\right]. \tag{2.29}$$

The controlled parameter of actual transmission lines under self-weight, $H$, is usually known. Considering the dynamic displacement of two-span transmission lines is measured from the static configuration under self-weight, first of all the static configuration should be obtained. The static configuration can be calculated by ABAQUS from the natural state of transmission lines. The natural state of transmission lines is a configuration without self-weight. Theoretically, any natural configuration of the first span with the length $L_{10}$ can be assumed to determine its static equilibrium configuration under self-weight by means of ABAQUS. If the natural state is assumed to be a catenary configuration, fewer iterations may be obtained to arrive at the static state with known quantity $H$. The natural configuration without self-weight can be assumed to be described with equation (2.26), in which the parameter $H$ is obtained by iteration solution of equation (2.28) after setting $L_1$ equal to $L_{10}$. It is noted that $H$ of the natural configuration may be understood as the parameter without the original physical definition, and the catenary curve is not the unique choice of the natural configuration of the first span. The static configuration is calculated by means of ABAQUS after application of gravity. Natural configurations of other spans can be obtained in the same way as the natural configuration of the first span.

The model of the typical transmission line section, which consists of two-span conductors and suspension insulator, is established to calculate the dynamic stiffness. The distances between the supports of the first and second spans are 200 m and 180 m, respectively. The inclination angles of first and second spans are the same as $10°$. The conductor type is aluminium conductor steel reinforced cable LGJ-400/50, mass per unit length is $1.511\,\mathrm{kg\,m}^{-1}$, Young's modulus $E$ is $7.0 \times 10^4\,\mathrm{MPa}$, and the dimension of the conductor is $27.63 \times 10^{-3}\,\mathrm{m}$. The suspension insulator string consists of 28 ball-and-socket porcelain insulators of model XP-16 with self-weight 6.0 kg, the length of the suspension insulator string is 6.47 m, and its Young's modulus is set to be 200 GPa. Twenty-eight truss elements are used to simulate the suspension insulator string. The horizontal tension of a span in the static configuration under self-weight is usually known and is $14.25\,\mathrm{kN}(\lambda_1 = 3\pi)$. According to the known horizontal tension in the static configuration and iteration solution of equation (2.28), the horizontal tensions used to determine the natural configuration are obtained, 16.69 kN of the first span and 17.47 kN of the second span. The static configuration is calculated by mean of ABAQUS under self-weight from the natural configuration, as shown in figure 3. To distinguish between the natural and static configurations, the displacement of static configuration has been enlarged 10 times in figure 3. The horizontal tension in the static configuration, which is calculated by ABAQUS, is 14.24 kN. The horizontal tension obtained by ABAQUS is very close to the setting value 14.25 kN. The first four modes and natural frequencies are shown in figure 3, and the dimensionless frequencies $\bar{\omega}_1/\pi$ are 1.065, 2.000, 2.221 and 2.598, respectively. As the shapes of the first and fourth modes in a span are symmetric, the modes are called the symmetric modes. The shape of the second mode in the first span is antisymmetric, and the displacement of the second mode in the second span is closed to zero. Therefore, the third and second modes are called antisymmetric modes.

Substituting the parameters of the previous example into equation (2.24), the dynamic stiffness, $K_d$, with respect to the excited frequency is obtained. The dynamic stiffness is normalized with respect to the corresponding static stiffness value to calculate the ordinates, as shown in figure 4. $K_s$ is the static stiffness. The static stiffness may be obtained from equation (2.24) by letting $\omega = 0$. The increment of the horizontal coordinate in the dynamic stiffness curve is 0.0005. The resonant peaks of the curves of the dynamic stiffness correspond to the dimensionless natural frequency of two-span transmission lines supported at both ends. The abscissa values corresponding to the first four peaks, which are, respectively, 1.063, 2.000, 2.222 and 2.595, are very close to the dimensionless natural frequencies obtained by ABAQUS. The comparison of natural frequencies obtained by two methods shows that the theoretical formula for dynamic stiffness is reasonable. With the dynamic stiffness curves in figure 4, it can be seen that the values of the first three peaks are much less than the value of the fourth peak. It is shown that the first three modes have a lower effect on conductor tension than the

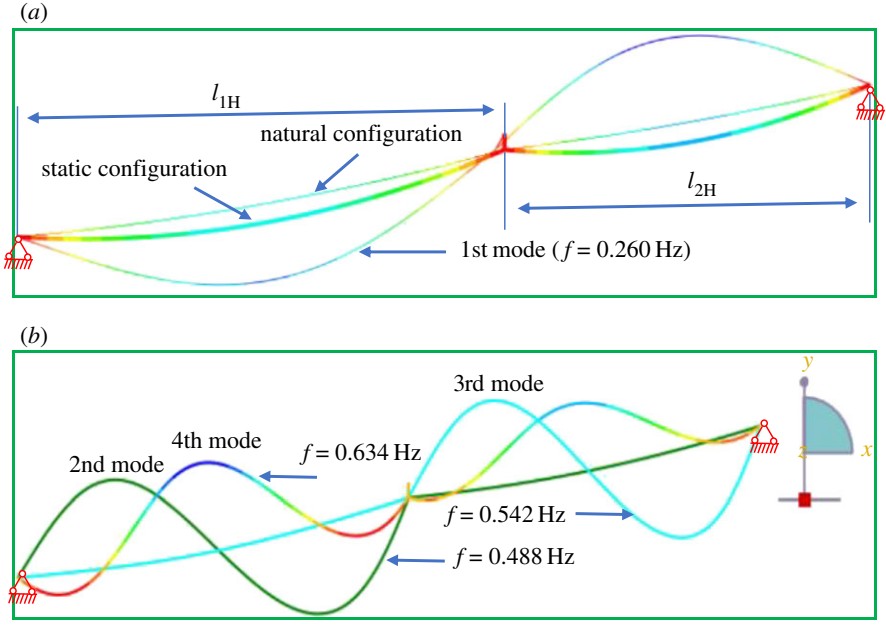

**Figure 3.** Different configurations of inclined cable and modes determined by ABAQUS. (*a*) Different configurations and the first mode. (*b*) Second, third and fourth modes.

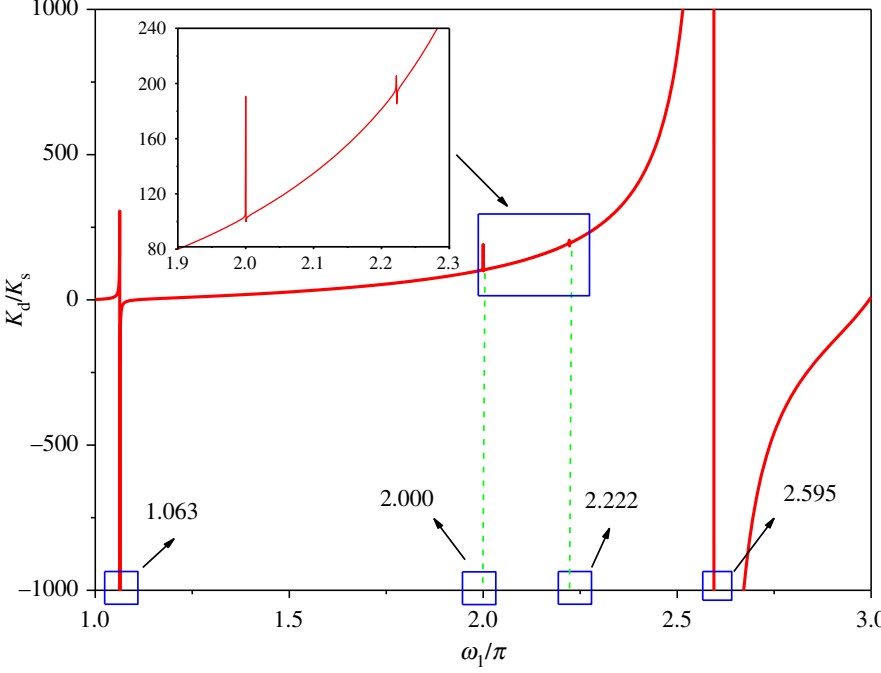

**Figure 4.** Variation of the dynamic stiffness with excited frequency.

fourth mode. The reason is that the values of the negative and positive peak for the first three mode shapes are close to each other, and the value of the positive peak for the fourth mode shape is much larger than that of the negative peak.

## 2.3. Effects of insulator string length on the dynamic stiffness

Insulator string is an important part of transmission lines and plays an important role in electrical insulation and mechanical support. The length of the insulator string depends on the voltage in transmission line design. In fact, the change in insulator string length will also have some effect on

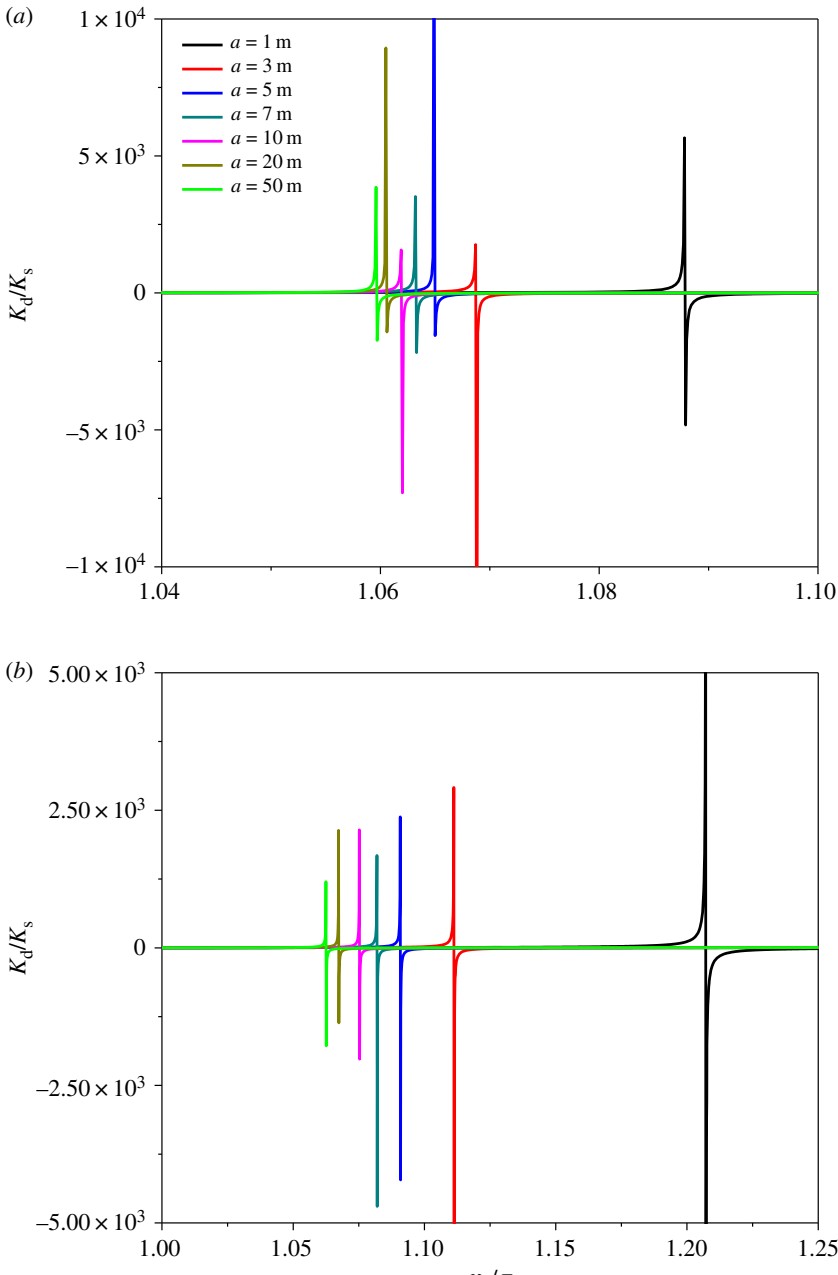

**Figure 5.** Effect of insulator string length on the dynamic stiffness under different geometrical-elastic parameters (200 – 180 m). (a) $\lambda_1/\pi = 3$. (b) $\lambda_1/\pi = 7$.

the mechanical characteristics of multi-span transmission lines. Therefore, the sensitivity of dynamic stiffness and natural frequency to insulator string length is investigated in the following. The same material parameters as the previous example are used. The span length of two-span transmission lines is 200 – 180 m. Increasing the insulator string length from 1 m to 50 m, the dynamic stiffness of two-span transmission lines with $\lambda_1/\pi = 3$ and $\lambda_1/\pi = 7$ is studied, respectively. When $\lambda_1$ is $3\pi$, the dynamic stiffness with different insulator string lengths is depicted in figure 5a. Compared with the results corresponding to other insulator strings, the value of the dimensionless natural frequency significantly decreases with increasing the insulator length from 1 to 3 m. It is shown that when insulator length is small, the effects of insulator string length on the natural frequency are all the more obvious. When $\lambda_1$ is $7\pi$, the dynamic stiffness with different string lengths is represented in figure 5b. The peak value on the dynamic stiffness curve decreases with increasing the insulator string length. The results also show that the natural frequency is more sensitive to the insulator string length when the insulator string length is small. The comparison of the two figures shows that the natural

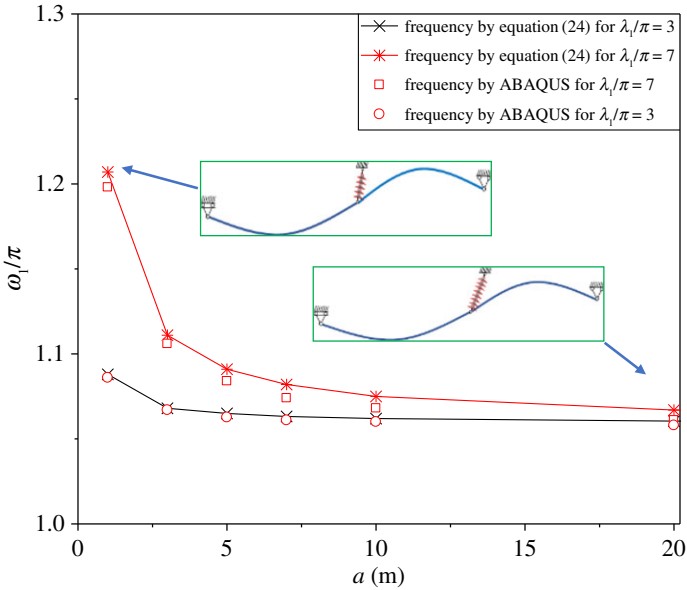

**Figure 6.** Variation of natural frequencies with insulator string length $a$ (200 − 180 m).

frequency of two-span transmission lines with $\lambda_1 = 7\pi$ is more sensitive to the insulator string length than that of two-span transmission lines with $\lambda_1 = 3\pi$.

In figure 6, the line symbols are the dimensionless natural frequencies corresponding to the first mode, which are identified by the dynamic stiffness peak in figure 5. In figure 6, the scatter plots are the dimensionless natural frequencies with different insulator string lengths, which are calculated by ABAQUS. The theoretical results are generally consistent with those by ABAQUS. The maximum error between theoretical and ABAQUS results is only 1% when insulator string length is 20 m and $\lambda_1/\pi = 7$. It can be seen that the dimensionless natural frequency decreases with increasing the insulator string length. The effect of the insulator string length on the natural frequency is greater when $\lambda_1/\pi = 7$ than that when $\lambda_1/\pi = 3$. Figure 6 also shows that the sensitivity of the natural frequency to the insulator string length is high when the insulator length is small. The natural frequency is close to a constant when the insulator length is greater than 5 m.

## 2.4. Effects of inclination angles on the dynamic stiffness

The inclination angle of transmission lines depends on the terrain. In order not to significantly increase the weight and top size of a transmission tower, the design inclination angle is generally small. Therefore, the transmission lines with the range of inclination angle as $-30° \sim 30°$ are used in this paper. Also, for convenience comparison, the inclination angle of the first span remains a constant value of $20°$ when the inclination angle of the second span is changed.

Let the span distances of first and second spans be 200 and 180 m, and $\lambda_1$ is $7\pi$. Figure 7 represents the variation of the dynamic stiffness with respect to the dimensionless frequency, for the two-span transmission lines having several different values of inclination angle $\beta_2$. As a result, the peak values of the dynamic stiffness depend on the inclination angle. The natural frequency, which is identified by the peak of the dynamic stiffness, decreases with the increasing inclination angle of the second span.

In order to further verify the correctness of the dynamic stiffness theory, the ABAQUS software is used to establish an FEM model of two-span transmission lines with the span length of 200−180 m and $\lambda_1$ of $7\pi$. The inclination angle $\beta_1$ is set to be $0° \sim 30°$, and $\beta_2$ is set to be $-30° \sim 30°$. Tables 1−4 show the in-plane natural frequencies of the first symmetrical mode calculated by ABAQUS and the theoretical dynamic stiffness, respectively. The maximum error between theoretical solution and ABAQUS results is found to be 4.23% when the inclination angles of the first and second spans are $0°$ and $30°$, respectively. The error is small when the inclination angle of the first span is equal to that of the second span. As is generally true of the natural frequency of the first symmetrical mode, the natural frequency decreases with increasing the inclination angle when $\beta_1$ is a constant.

Assume the span lengths of two-span are 200 and 180 m, the inclination angles of the first and second spans are equal to $20°$. In figure 8, the solid lines represent the variation of natural frequencies corresponding to the first four modes with the geometric parameter $\lambda_1/\pi$, the scatter plots are the

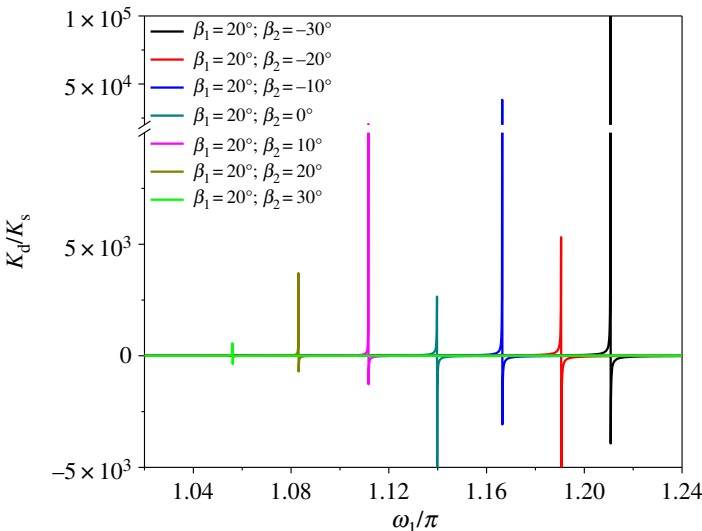

**Figure 7.** Effect of the inclination angle on the dynamic stiffness.

**Table 1.** Natural frequency of the first symmetrical mode when $\beta_1 = 0°$.

| $\beta_1(°)$ | $\beta_2(°)$ | theoretical solution $\bar{\omega}_1/\pi$ | ABAQUS $\bar{\omega}_1/\pi$ | error (%) |
|---|---|---|---|---|
| 0 | −30 | 1.158 | 1.202 | 3.66 |
| 0 | −20 | 1.134 | 1.146 | 1.05 |
| 0 | −10 | 1.108 | 1.105 | 0.27 |
| 0 | 0 | 1.083 | 1.079 | 0.37 |
| 0 | 10 | 1.059 | 1.057 | 0.19 |
| 0 | 20 | 1.038 | 1.051 | 1.24 |
| 0 | 30 | 1.020 | 1.065 | 4.23 |

natural frequencies identified by the peak of the dynamic stiffness. The results show that the in-plane natural frequencies calculated by the dynamic stiffness theory coincide well with the results obtained by the ABAQUS.

In figure 8, the first curve starts with a unit ordinate and increases with increasing geometrical parameter $\lambda_1/\pi$. As can be seen from equation (2.24), the first natural frequency is close to 1 when the tension of the conductor gradually increases. The modal shape corresponding to the first curve is shown in figure 9a. Figure 9a shows that the mode displacement is zero over the entire second span length when $\lambda_1/\pi = 0.1$. The swing angle of the insulator string gradually decreases with increasing the tension of the conductor. The insulator string has the largest swing angle when $\lambda_1/\pi = 10$. The phenomenon reveals that the effect between two spans in the first modal shape of two-span transmission lines decreases with increasing the horizontal tension.

In figure 8, the value of the second natural frequency curve starts at an ordinate of 1.11 and increases with increasing geometrical parameter $\lambda_1/\pi$. When the second natural frequency comes increasingly closer to the third natural frequency, there is a transition region where the two frequencies are close to each other but distinct as shown in subgraph 1 of figure 8. It is shown that the transition region is very small, and the plot is very similar to that obtained by the dynamic stiffness theory. Then, the corresponding mode changes from symmetrical mode or antisymmetrical mode to a hybrid mode, a mixture of symmetrical and antisymmetric forms as shown in figure 9b. The transition of modal shapes happens gradually as the geometrical parameter $\lambda_1/\pi$ from 1.8 to 2. The mixture shape of the hybrid mode is mainly confined to the first span. When the geometrical parameter $\lambda_1/\pi$ is larger than 2, the second natural frequency is independent of the $\lambda_1/\pi$, and it is approximately at a constant of 2. Based on a single-span inclined cable, the transition region and hybrid modes have been studied by Triantafullou [6], Wu [7] and Lai et al. [8].

**Table 2.** Natural frequency of the first symmetrical mode when $\beta_1 = 10°$.

| $\beta_1(°)$ | $\beta_2(°)$ | theoretical solution $\bar{\omega}_1/\pi$ | ABAQUS $\bar{\omega}_1/\pi$ | error (%) |
|---|---|---|---|---|
| 10 | −30 | 1.181 | 1.220 | 3.20 |
| 10 | −20 | 1.159 | 1.166 | 0.60 |
| 10 | −10 | 1.134 | 1.128 | 0.53 |
| 10 | 0 | 1.109 | 1.095 | 1.28 |
| 10 | 10 | 1.083 | 1.076 | 0.65 |
| 10 | 20 | 1.059 | 1.068 | 0.84 |
| 10 | 30 | 1.037 | 1.077 | 3.71 |

**Table 3.** Natural frequency of the first symmetrical mode when $\beta_1 = 20°$.

| $\beta_1(°)$ | $\beta_2(°)$ | theoretical solution $\bar{\omega}_1/\pi$ | ABAQUS $\bar{\omega}_1/\pi$ | error (%) |
|---|---|---|---|---|
| 20 | −30 | 1.211 | 1.234 | 1.86 |
| 20 | −20 | 1.191 | 1.184 | 0.59 |
| 20 | −10 | 1.166 | 1.145 | 1.83 |
| 20 | 0 | 1.140 | 1.114 | 2.33 |
| 20 | 10 | 1.112 | 1.091 | 1.92 |
| 20 | 20 | 1.083 | 1.078 | 0.46 |
| 20 | 30 | 1.056 | 1.081 | 2.31 |

**Table 4.** Natural frequency of the first symmetrical mode when $\beta_1 = 30°$.

| $\beta_1(°)$ | $\beta_2(°)$ | theoretical solution $\bar{\omega}_1/\pi$ | ABAQUS $\bar{\omega}_1/\pi$ | error (%) |
|---|---|---|---|---|
| 30 | −30 | 1.257 | 1.252 | 0.40 |
| 30 | −20 | 1.238 | 1.206 | 2.65 |
| 30 | −10 | 1.213 | 1.221 | 0.63 |
| 30 | 0 | 1.185 | 1.203 | 1.50 |
| 30 | 10 | 1.153 | 1.169 | 1.37 |
| 30 | 20 | 1.118 | 1.090 | 2.57 |
| 30 | 30 | 1.083 | 1.080 | 0.28 |

In figure 8, the third natural frequency curve starts at an ordinate of 2 and passes the first transition region. When $\lambda_1/\pi$ is in the range of $2 \sim 2.2$, the third natural frequency curve increases with increasing $\lambda_1/\pi$. When the third natural frequency comes increasingly closer to the fourth natural frequency, there is the second transition region where the two frequencies are close to each other but distinct as shown in subgraph 2 of figure 8. The second transition region is also very small and this plot is very similar to that identified by the dynamic stiffness. In the second transition region, the mode corresponding to the third natural frequency is the hybrid mode. The hybrid mode as shown in figure 9c gradually transforms into an antisymmetrical mode when the geometrical parameter $\lambda_1/\pi$ increases from 2.2 to 2.4. When the geometrical parameter $\lambda_1/\pi$ increases from 2.2 to 2.4 in the second transition region, the antisymmetrical mode, which corresponds to the fourth natural frequency, changes into hybrid mode. Then this hybrid mode as shown in figure 9d gradually changes into the symmetrical mode. The shape of the hybrid mode is mainly confined to the second span. The results demonstrate that the

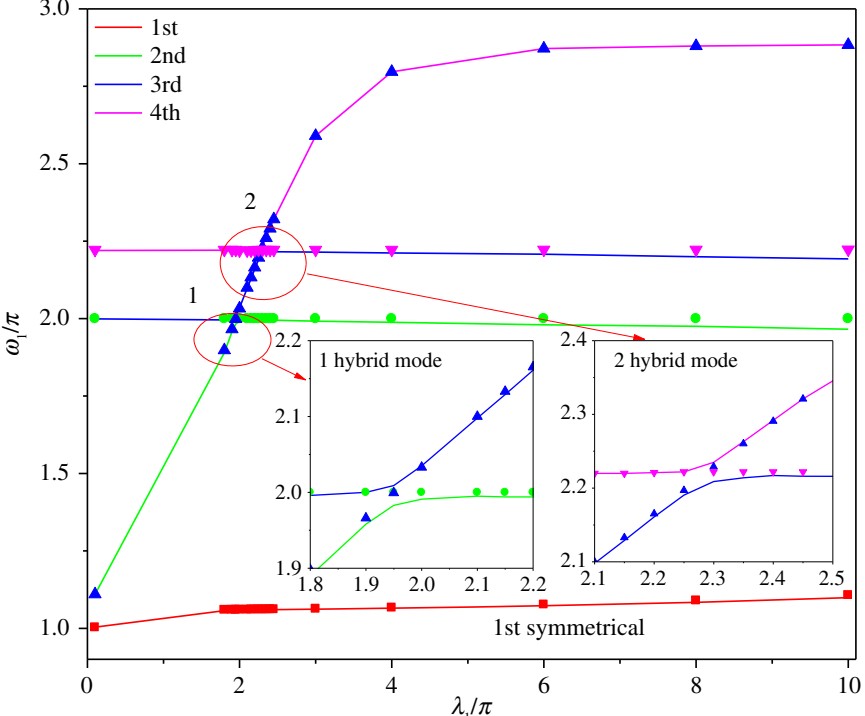

**Figure 8.** In-plane natural frequencies of two-span transmission lines ($20° \sim 20°$).

dynamic stiffness of two-span transmission lines, in which a span is attached at different heights, offers a simple yet accurate means of expressing the natural frequency.

## 2.5. Effects of inclination angle on modal shape

The modal shape is calculated by equation (2.16a,b). It can be seen that the modal shape is asymmetrical. This is because the displacements at both ends of a span are not zero. Figure 9 shows that the modal shapes are a mixture of symmetrical shape and antisymmetrical shape when the corresponding natural frequency is in the transition region. The modal shapes are symmetrical or antisymmetrical when the corresponding natural frequency is in other regions. Considering the transition region is very small, the asymmetrical mode is divided into two parts, symmetrical shape and antisymmetrical shape. The asymmetrical modal function is simplified. When the natural frequency is in other regions, the corresponding symmetrical modal shape is given by

$$\phi_{si}(x) = \frac{1}{D_i} \begin{cases} \dfrac{\cos\beta_1}{\bar{\omega}_1^2 - A_1\lambda_1^2} \dfrac{q_{y_1}l_1}{H_1^2}\left(1 - \sin\dfrac{\bar{\omega}_1}{l_1}x_1\tan\dfrac{\bar{\omega}_1}{2} - \cos\dfrac{\bar{\omega}_1}{l_1}x_1\right) \\ -\dfrac{\cos\beta_2}{\bar{\omega}_2^2 - A_2\lambda_2^2} \dfrac{q_{y_2}l_2}{H_2^2}\left(1 - \sin\dfrac{\bar{\omega}_2}{l_2}x_2\tan\dfrac{\bar{\omega}_2}{2} - \cos\dfrac{\bar{\omega}_2}{l_2}x_2\right) \end{cases}, \tag{2.30}$$

where $i = 1,2,3,\ldots$, signify the first, second, third, etc. symmetrical in-plan modes, respectively. The modal function is normalized with respect to $D_i$. $D_i$ is the maximum absolute value of modal displacement. The antisymmetrical mode is given by

$$\phi_{ai}(x) = \begin{cases} 0 \\ \left(\dfrac{x_2}{l_2} - \dfrac{\sin(\bar{\omega}_2/l_2)x_2}{\sin\bar{\omega}_2}\right) & (0 \le x_2 \le l_2) \end{cases} \tag{2.31a}$$

and

$$\phi_{ai}(x) = \begin{cases} \left(\dfrac{x_1}{l_1} - \dfrac{\sin(\bar{\omega}_1/l_1)x_1}{\sin\bar{\omega}_1}\right) & (0 \le x_1 \le l_1) \\ 0 \end{cases}, \tag{2.31b}$$

where $i = 1,2,3,\ldots$, signify the first, second, third, etc. antisymmetrical in-plan modes, respectively. The equation (2.31a,b) is valid when $\bar{\omega}_1$ or $\bar{\omega}_2$ is close to $2\pi$.

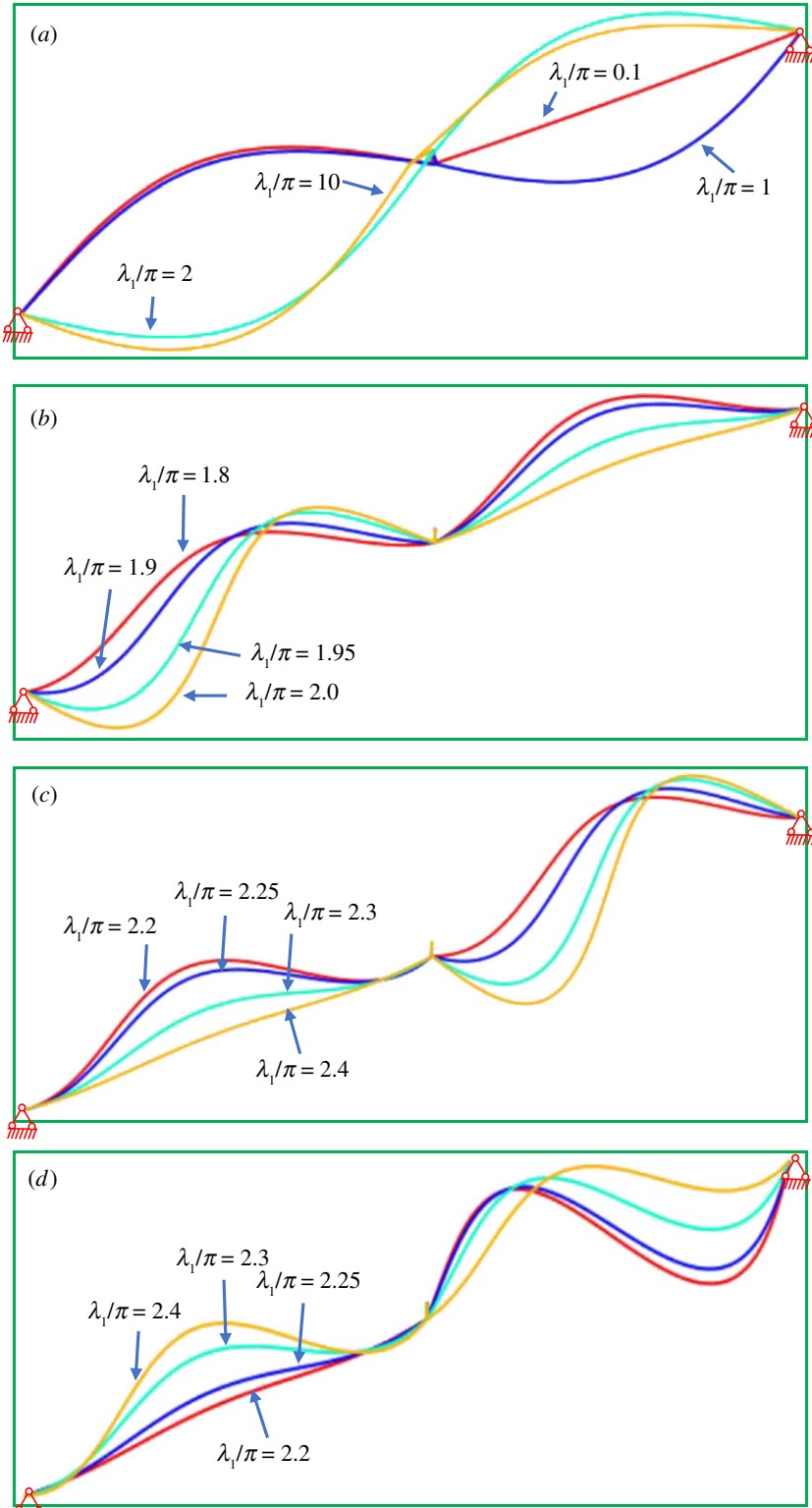

**Figure 9.** First fourth modal shapes of two-span transmission lines with different geometric parameters. (*a*) First mode. (*b*) Second mode. (*c*) Third mode. (*d*) Fourth mode.

In order to verify the simplified modes presented in equations (2.30) and (2.31*a,b*), the FEM model of two-span transmission lines, in which the span lengths are 200–180 m and the inclination angles are $-20°$ to $-20°$, have been established in ABAQUS. Considering the restrictions regarding the application of the theoretical formulae, the geometrical parameter $\lambda_1/\pi$ is set to 2.5. At the outside of the transition region, all modes exhibit the approximate symmetrical or antisymmetrical modal shape in a span. As presented in figure 10*a*, the solid lines are the first two symmetrical modes calculated by

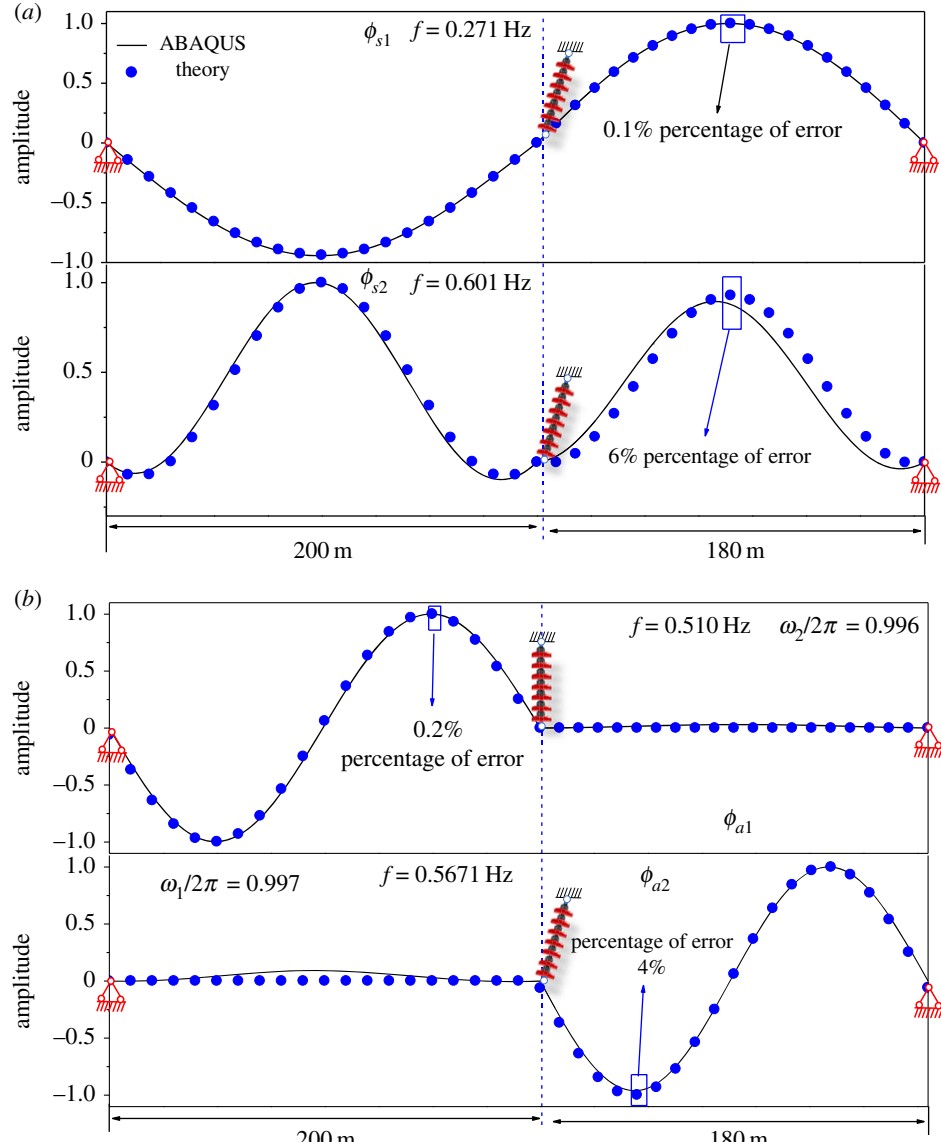

**Figure 10.** First four natural modal shapes of two-span with $\lambda_1/\pi = 2.5$. (*a*) Symmetrical modal shapes ($\phi_{s1}$, $\phi_{s2}$). (*b*) Antisymmetrical modal shapes ($\phi_{a1}$, $\phi_{a2}$).

ABAQUS, and the scatter plots are calculated by equation (2.30). The first symmetrical mode calculated by equation (2.30) coincides well with the FEM result obtained by ABAQUS. As shown in figure 10*b*, the first antisymmetrical modes calculated by Equation (2.31*b*) and ABAQUS are in good agreement. For the second symmetrical mode and the second antisymmetrical mode, the modal shape calculated by the analytic solution is different from that calculated by ABAQUS, as shown in figure 10. The reason for this error is that a mode interacts with another mode when the natural frequencies corresponding to these two modes are close to each other. The interaction between the two modes is to weaken with the increase of the geometrical parameter $\lambda_1/\pi$. The errors between the two modal shapes calculated by the theoretical expression and ABAQUS decrease.

From figure 9 and equation (2.16*a,b*), it can be seen that when $\lambda_1/\pi$ is close to 2.3, the natural frequencies corresponding to the third and fourth modes are close to $2\pi$. When the two natural frequencies are close to each other, the symmetrical mode calculated by equation (2.30) and the antisymmetrical mode calculated by equation (2.31*a,b*) are orthogonal. Therefore, according to the linear superposition principle, the hybrid modes are given by the symmetrical mode and the antisymmetrical mode as

$$\phi_{h(i+k+m-2)}(x) = b_{mi}\phi_{si}(x) + d_{mk}\phi_{ak}(x), \tag{2.32}$$

where $m = 1,2$, signifies the span number.

In order to verify the applicability of equation (2.32), in the following analysis, the span lengths (200–180 m) and the inclination angles (20°–20°) are fixed and the geometrical parameter $\lambda_1/\pi$ is changed gradually. The hybrid modes of the two-span with the geometrical parameter $\lambda_1/\pi = 2.4$, $\lambda_1/\pi = 2.3$ and $\lambda_1/\pi = 2.25$ are calculated, respectively, by equation (2.32) and the FEM. As shown in figure 11, the solid lines are hybrid modes obtained by ABAQUS. In figure 11, it can be found that when the mode is changed from the symmetrical shape or antisymmetric shape to hybrid modes, the corresponding dimensionless natural frequency $\bar{\omega}_2$ is very close to $2\pi$, which means that it is reasonable to use the equation (2.32) to describe the hybrid modal shape. In figure 11, the scatter plots are hybrid modes, which are calculated by equation (2.32). It is clear that the modes calculated by theoretical method and FEM are in good agreement. The results indicate that the simplified theoretical formula can describe hybrid modes better for two-span transmission lines with small inclination angles.

## 2.6. Effects of damping on dynamic stiffness and modal shape

In the previous section, only the modal shapes and natural frequencies were discussed. In many cases, the influence of damping upon the response of a vibratory system is minor and can be disregarded. In general, since the effects are not known in advance, damping must be considered in the vibration analysis of any system [10]. In order to study the effects of damping on the modal shape and dynamic stiffness, the equation (2.15a,b) may be written in the form

$$\frac{d^2\bar{w}_{t1}}{d\zeta^2} + \left(\left(\frac{\omega}{\omega_{01}}\right)^2 - i2\xi_1\frac{\omega}{\omega_{01}}\right)\bar{w}_{t1} = \frac{q_{y_1}\bar{h}_{\tau1}l_1^2}{H_1^2} - \bar{\omega}_1^2\Delta\bar{x}_1\sin\beta_1 - \bar{\omega}_1^2(\Delta\bar{x}_2\sin\beta_1 - \Delta\bar{x}_1\sin\beta_1)\zeta \qquad (2.33a)$$

and

$$\frac{d^2\bar{w}_{t2}}{d\zeta^2} + \left(\left(\frac{\omega}{\omega_{02}}\right)^2 - i2\xi_2\frac{\omega}{\omega_{02}}\right)\bar{w}_{t2} = \frac{q_{y_2}\bar{h}_{\tau2}l_2^2}{H_2^2} - \bar{\omega}_2^2\Delta\bar{x}_2\sin\beta_2 + \bar{\omega}_2^2\Delta\bar{x}_2\zeta\sin\beta_2, \qquad (2.33b)$$

where

$$\omega_{01} = \frac{1}{l_1}\sqrt{\frac{H_1}{m}}; \quad \omega_{02} = \frac{1}{l_2}\sqrt{\frac{H_2}{m}}; \quad \xi_1 = \frac{c}{2m\omega_{01}}; \quad \xi_2 = \frac{c}{2m\omega_{02}}.$$

For simplicity, $\xi_1$ and $\xi_2$ are defined. $\xi_1$ and $\xi_2$ are the damping parameters of the first and second spans, respectively. In transmission lines, internal damping mainly comes from the conductor cables. When a dynamic load is applied to a conductor, the internal damping arises from axial friction between the strands. As a matter of act, it is difficult to accurately determine the damping ratio of the conductor. Different values of viscous damping ratios in the range of $0.08 \sim 4.0\%$ for vertical movement are used in the numerical study [32,41,45].

In order to study the effects of damping on the modal shape and natural frequency, the model of two-span transmission lines, in which the span lengths are 200–180 m and the inclination angles are 20°–20°, have been established. The geometrical parameter $\lambda_1/\pi$ is set to 2.5. Next, the effect of the dimensionless factor $\xi_1$ on dynamic stiffness was studied. Figure 12a shows the real part of dynamic stiffness, and figure 12b shows the corresponding imaginary part of dynamic stiffness. As would be expected, damping significantly affects the dynamic stiffness only at excited frequencies close to the natural frequencies of two-span transmission lines.

Figure 13a shows the real and imaginary components of the first complex mode. It is shown that the damping parameter significantly affects the shape of the real component of the first complex mode when the value of the damping parameter is large. It is important to note that the shape of the real component of the first complex mode is different from the shape of the imaginary component. Figure 13b shows the real and imaginary components of the second complex mode. It is shown that the damping parameter does not significantly affect the shape of the real component of the second complex mode. The shape of the real component of the second complex mode is close to that of the imaginary component. Figure 13c shows the real and imaginary components of the first antisymmetrical mode. The damping parameter does not alter the shape of the real component of the first antisymmetrical mode. In addition, the shape of the real component of the first antisymmetrical mode is the same as that of the imaginary component.

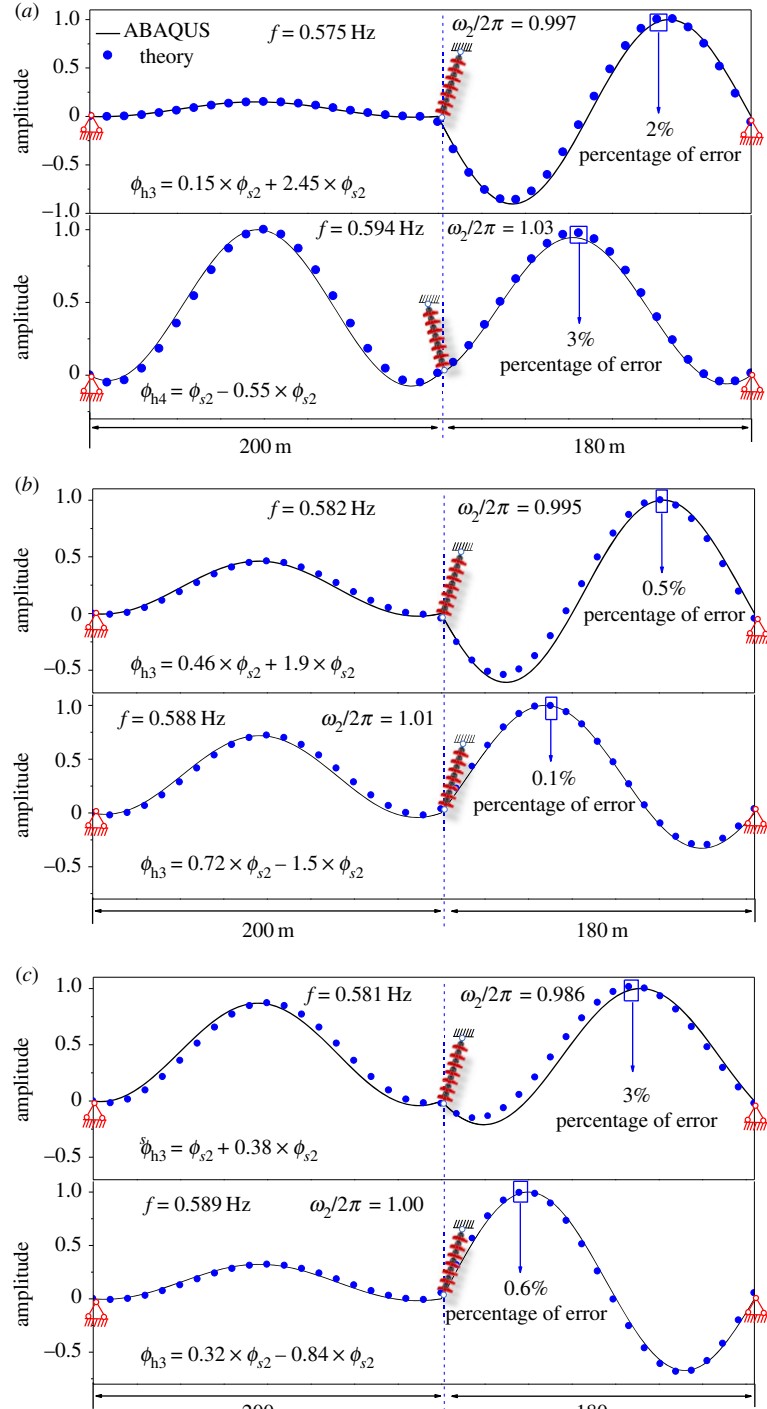

**Figure 11.** Hybrid modal shape of two-span with different geometrical parameters. (*a*) Hybrid mode of two-span with $\lambda_1/\pi = 2.4$. (*b*) Hybrid mode of two-span with $\lambda_1/\pi = 2.3$. (*c*) Hybrid mode of two-span with $\lambda_1/\pi = 2.25$.

## 3. Dynamic stiffness and modal shape of multi-span transmission lines

Similarly, according to the method for calculating dynamic stiffness of a two-span transmission line, the dynamic stiffness model of a multi-span transmission line with an arbitrary number of spans and inclination angles is established, as shown in figure 14. This multi-span transmission line consists of $N$ spans and $N - 1$ insulator strings. In addition, in this series of spans between dead-ends, adjacent spans may differ in length, or their supports may be at different elevations. The left end $A_1$ that is attached to a smooth roller may be moving along the horizontal direction. The right end $A_{N+1}$ is a fixed hinge. Every span as a substructure has its own local coordinate system, which is attached to the

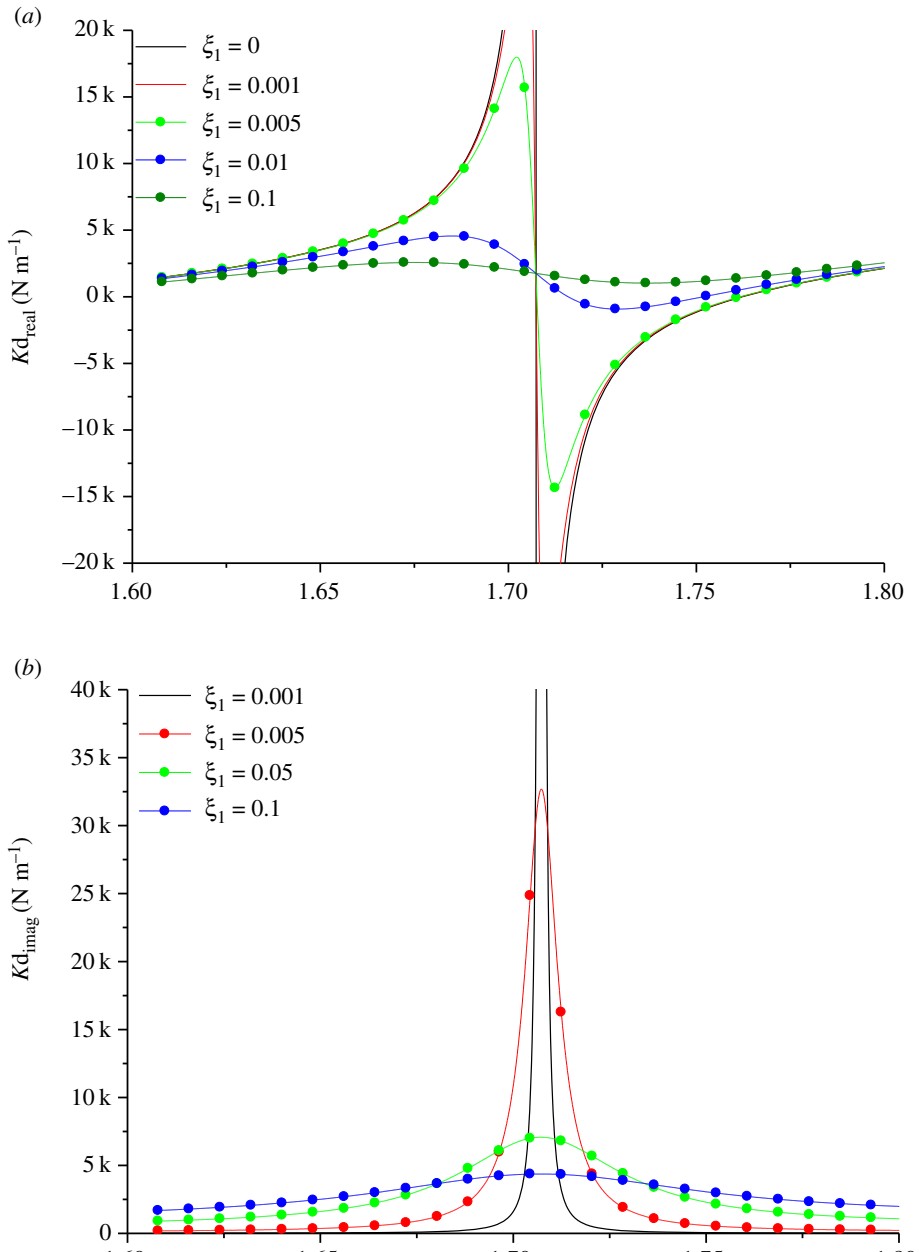

**Figure 12.** Dynamic stiffness of damped two-span transmission lines. (*a*) Real component. (*b*) Imaginary component.

left end of each span. For *i*-th span, let $x_i A_i y_i$ be a coordinate system, with the origin taken at the left end $A_i$, the $x_i$-axis taken along the cable chord, and the $y_i$-axis taken along the normal direction. The dynamic displacements of *i*-th span, $u_i(x,t)$ and $w_i(x_i,t)$, are measured from the position of its static equilibrium in directions parallel to the $x_i$-axis and $y_i$-axis, respectively.

For the multi-span transmission lines associated with the position of static equilibrium, the left end $A_1$ is subjected to a force $F$ in the horizontal direction. If the left end $A_1$ is subjected to an additional harmonically varying horizontal force $\Delta F = \Delta\bar{F}e^{i\omega t}$, under the action of the force, the multi-span transmission lines will have small-amplitude displacements in the neighbourhood of the static equilibrium position as the dotted line in figure 14. Neglecting the conductor inertia component parallel to the chord, the displacement in the normal direction for *i*-th span conductor with respect to the static equilibrium position can be governed by the differential equations

$$H_i \frac{\partial^2 w_i}{\partial x_i^2} + h_{\tau i} \frac{d^2 y_i}{dx_i^2} = m \frac{\partial^2 w_i}{\partial t^2} + c \frac{\partial w_i}{\partial t}, \tag{3.1}$$

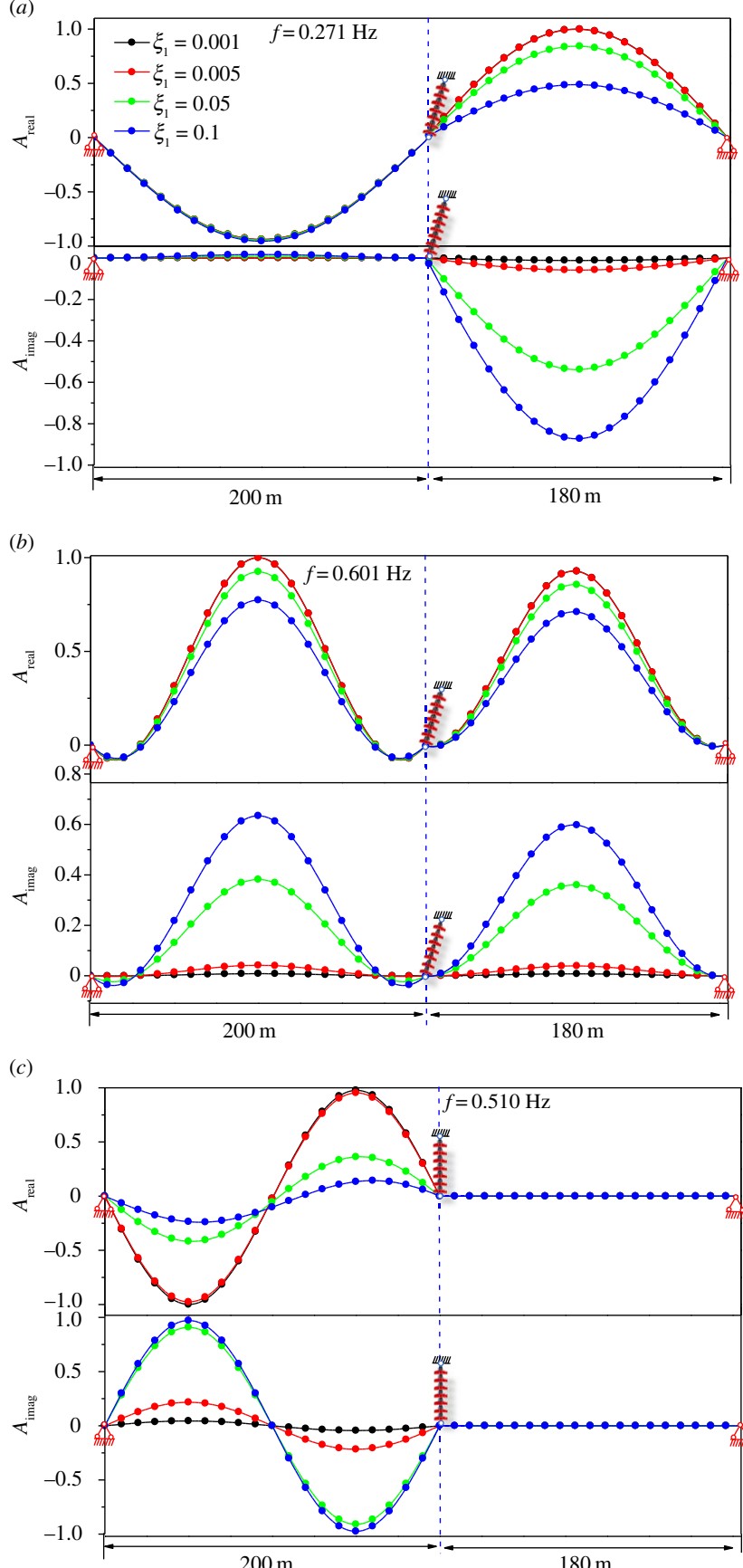

**Figure 13.** Modal shape for damped transmission lines. (*a*) Modal shape of the first symmetrical mode. (*b*) Modal shape of the second symmetrical mode. (*c*) Modal shape of the first antisymmetrical mode.

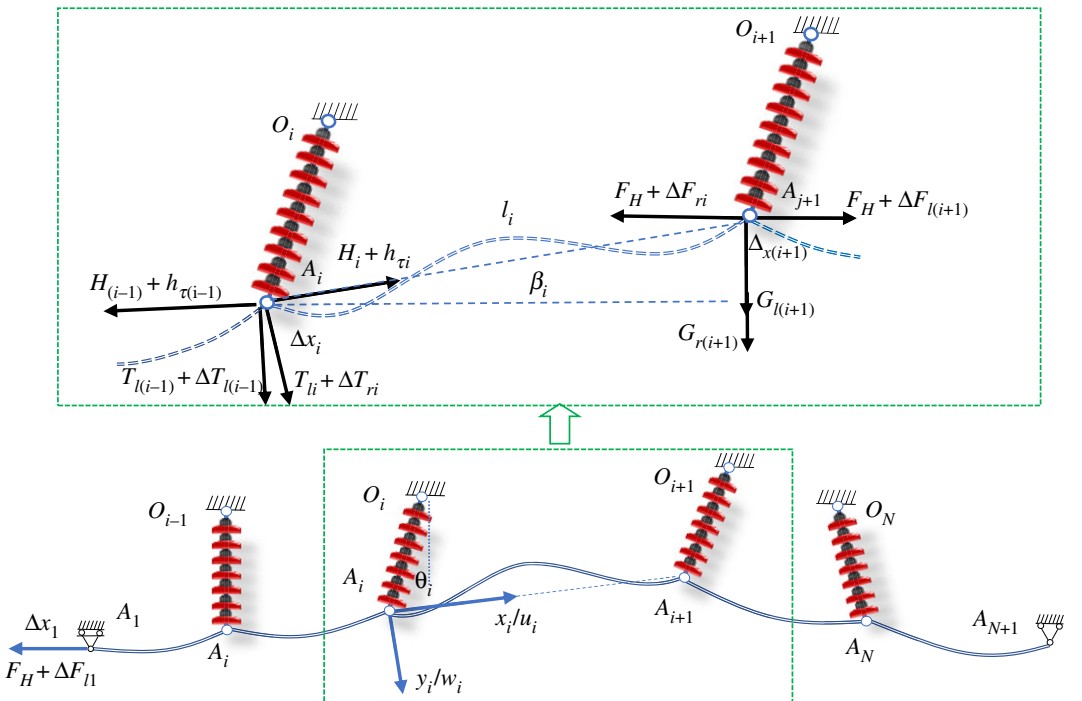

**Figure 14.** Multi-span transmission lines.

in which the $h_{\tau i}$ is the additional tension in $H_i$. $y_i$ is the static equilibrium configuration of the $i$-th span conductor represented by equation (2.1). The solution to the differential equation can be obtained using the technique known as separation of variables. When point $A_1$ in multi-span transmission lines is subjected to a harmonic force, the increment displacements and additional tensions of the $i$-th span are given by

$$\Delta F(t) = \Delta \bar{F} e^{i\omega t}; \quad h_{\tau i}(t) = \bar{h}_{\tau i} e^{i\omega t}; \quad w_{ti}(x,t) = \bar{w}_{ti}(x) e^{i\omega t} \tag{3.2a}$$

and

$$\Delta x_i(t) = \Delta \bar{x}_i e^{i\omega t}; \quad \Delta x_{i+1}(t) = \Delta \bar{x}_{i+1} e^{i\omega t}. \tag{3.2b}$$

Using equation (3.2a,b), then equation (3.1) becomes

$$\frac{d^2 \bar{w}_i}{d\zeta^2} + \bar{\omega}_i^2 \bar{w}_i = \frac{q_{y_i} \bar{h}_{\tau i} l_i^2}{H^2}, \tag{3.3}$$

where $\bar{\omega}_i^2 = (m\omega^2 - \omega c i) l_i^2 / H_i$, $\zeta = x_i / l_i$. The boundary conditions at points $A_i$ and $A_{i+1}$ are

$$w_i(0) = \Delta x_i \sin\beta_i; \quad w_i(l_i) = \Delta x_{i+1} \sin\beta_i. \tag{3.4}$$

The solutions of equation (3.3), with the given boundary conditions of equation (3.4), are

$$\bar{w}_i(x) = \frac{q_{y_i} \bar{h}_{\tau i} l_i^2}{\bar{\omega}_i^2 H_i^2} \left(1 - \sin\bar{\omega}_i\zeta \tan\frac{\bar{\omega}_i}{2} - \cos\bar{\omega}_i\zeta\right) + \Delta\bar{x}_{i+1}\sin\beta_i \frac{\sin\bar{\omega}_i\zeta}{\sin\bar{\omega}_1} + \Delta\bar{x}_i \sin\beta_i \left(\cos\bar{\omega}_i\zeta - \frac{\sin\bar{\omega}_i\zeta}{\tan\bar{\omega}_i}\right), \tag{3.5}$$

where equation (3.5) represents the modal shape of the $i$-th span. Substituting equation (3.5) into equation (2.3a,b), the tension increment in $H$ is given by

$$\bar{h}_{\tau i} = \frac{AE}{l_i}\cos\beta_i \frac{(\Delta\bar{x}_{i+1} - \Delta\bar{x}_i) - (q_{yi}l_i/2H_i)A_i(\Delta\bar{x}_i + \Delta\bar{x}_{i+1})\tan\beta_i}{1 - (\lambda_i^2/\bar{\omega}_i^2)A_i}, \tag{3.6}$$

where

$$\lambda_i^2 = \frac{EA}{H_i}\left(\frac{q_{y_i}l_i}{H_i}\right)^2; \quad A_i = 1 - 2\frac{\tan(\bar{\omega}_i/2)}{\bar{\omega}_i}. \tag{3.7}$$

The tension of the $i$-th span can be divided into a normal component, $\Delta T_i$, and a component parallel to the chord joining two support points, $H_i$. $\Delta T_{li}$ and $\Delta T_{ri}$ are normal components of the tension acting at two

support points of the $i$-th span. Using the separation of variables, the normal components are given by

$$\Delta \bar{T}_{li} = \bar{h}_{\tau i} \frac{q_{yi} l_i}{2 H_i} A_i + H_i \sin \beta_i \left( \frac{\Delta \bar{x}_{i+1}}{l_i} \cdot \frac{\bar{\omega}_i}{\sin \bar{\omega}_i} - \frac{\Delta \bar{x}_i}{l_i} \frac{\bar{\omega}_i}{\tan \bar{\omega}_i} \right) \tag{3.8a}$$

and

$$\Delta \bar{T}_{ri} = -\bar{h}_{\tau i} \frac{q_{yi} l_i}{2 H_i} A_i + H_i \sin \beta_i \left( \frac{\Delta \bar{x}_{i+1}}{l_i} \cdot \frac{\bar{\omega}_i}{\tan \bar{\omega}_i} - \frac{\Delta \bar{x}_i}{l_i} \frac{\bar{\omega}_i}{\sin \bar{\omega}_1} \right). \tag{3.8b}$$

The horizontal components of conductor tension acting at $A_i$, $\Delta F_{r(i-1)}$ and $\Delta F_{li}$ are given by

$$\Delta F_{r(i-1)} = \bar{h}_{\tau(i-1)} \cos \beta_{i-1} + \Delta T_r(i-1) \sin \beta_{i-1} \tag{3.9a}$$

and

$$\Delta F_{li} = \bar{h}_{\tau i} \cos \beta_i + \Delta T_{li} \sin \beta_i. \tag{3.9b}$$

The vertical components of conductor tension acting at $A_i$, $G_{r(i-1)}$ and $G_{li}$ are given by

$$G_{r(i-1)} = \frac{q}{2} l_{i-1} + H_{i-1} \tan \beta_{i-1}; \quad G_{li} = \frac{q}{2} l_i - H_i \tan \beta_i. \tag{3.10}$$

Assuming the swing angle $\theta_i$ of the insulator string is small, the moment equation of the insulator string around the suspension point $O_i$ is obtained as

$$a(\Delta F_{li} - \Delta F_{r(i-1)}) = J \ddot{\theta}_i + (G_{r(i-1)} + G_{li}) \Delta x_i + W \frac{\Delta x_i}{2}. \tag{3.11}$$

Using the separation of variables, substituting equations (3.9a,b) and (3.10) into equation (3.11) gives,

$$\Delta \bar{F} = B_{12}^1 \Delta \bar{x}_2 - B_{11}^1 \Delta \bar{x}_1 \quad (i = 1), \tag{3.12a}$$

$$B^i \Delta \bar{x}_i = B_{12}^i \Delta \bar{x}_{i+1} + B_{21}^{i-1} \Delta \bar{x}_{i-1} \quad (1 < i < N) \tag{3.12b}$$

and

$$B^N \Delta \bar{x}_N - B_{21}^{N-1} \Delta \bar{x}_{N-1} = 0 \quad (i = N), \tag{3.12c}$$

where

$$B_{21}^{i-1} = \frac{(AE/l_{i-1}) \cos^2 \beta_{i-1} (1 - ((q_{y(i-1)} l_{i-1}/2 H_{i-1}) A_{i-1} \tan \beta_{i-1})^2)}{1 - (\lambda_{i-1}^2/\bar{\omega}_{i-1}^2) A_{i-1}} + \frac{H_{i-1} \sin^2 \beta_{i-1}}{l_{i-1}} \frac{\bar{\omega}_{i-1}}{\sin \bar{\omega}_{i-1}}, \tag{3.13a}$$

$$B_{22}^{i-1} = \frac{(AE/l_{i-1}) \cos^2 \beta_{i-1} (1 - (q_{y(i-1)} l_{i-1}/2 H_{i-1}) A_{i-1} \tan \beta_{i-1})^2}{1 - (\lambda_{i-1}^2/\bar{\omega}_{i-1}^2) A_{i-1}} + \frac{H_{i-1} \sin^2 \beta_{i-1}}{l_{i-1}} \cdot \frac{\bar{\omega}_{i-1}}{\tan \bar{\omega}_{i-1}}, \tag{3.13b}$$

$$B_{11}^i = \frac{(AE/l_i) \cos^2 \beta_i ((q_{yi} l_i/2 H_i) A_i \tan \beta_i + 1)^2}{1 - (\lambda_i^2/\bar{\omega}_i^2) A_i} + \frac{H_i \sin^2 \beta_i}{l_i} \frac{\bar{\omega}_i}{\tan \bar{\omega}_i}, \tag{3.13c}$$

$$B_{12}^i = \frac{AE}{l_i} \cos^2 \beta_i \frac{(1 - ((q_{yi} l_i/2 H_i) A_i \tan \beta_i)^2)}{1 - (\lambda_i^2/\bar{\omega}_i^2) A_i} + \frac{H_i \sin^2 \beta_i}{l_i} \cdot \frac{\bar{\omega}_i}{\sin \bar{\omega}_i} \tag{3.13d}$$

and

$$B^i = B_{11}^i + B_{22}^{i-1} + \left( -\frac{\omega^2 J}{a^2} + \frac{(G_{r(i-1)} + G_{li})}{a} + \frac{W}{2a} \right). \tag{3.13e}$$

Finally, combining equation (3.12a−c) and $K_d = \underset{\Delta x_1 \to 0}{Lim} \frac{\Delta F}{\Delta x_1}$, the dynamic stiffness of multi-span transmission lines is given by

$$\begin{vmatrix} K_x + B_{11}^1 & -B_{12}^1 & 0 & 0 & 0 & 0 & \cdots \\ -B_{21}^1 & B^2 & -B_{12}^2 & 0 & 0 & 0 & \cdots \\ 0 & \ddots & \ddots & \ddots & \ddots & \ddots & \cdots \\ \vdots & 0 & -B_{21}^{i-1} & B^i & -B_{12}^i & 0 & \cdots \\ 0 & \cdots & 0 & \ddots & \ddots & \ddots & \cdots \\ \vdots & \vdots & \vdots & 0 & -B_{21}^{N-1} & B^{N-1} & -B_{12}^N \\ 0 & 0 & \cdots & \cdots & \cdots & -B_{21}^{N-1} & B^N \end{vmatrix} = 0 \tag{3.14}$$

An important formulation is obtained to calculate the dynamic stiffness of transmission lines with an arbitrary number of spans.

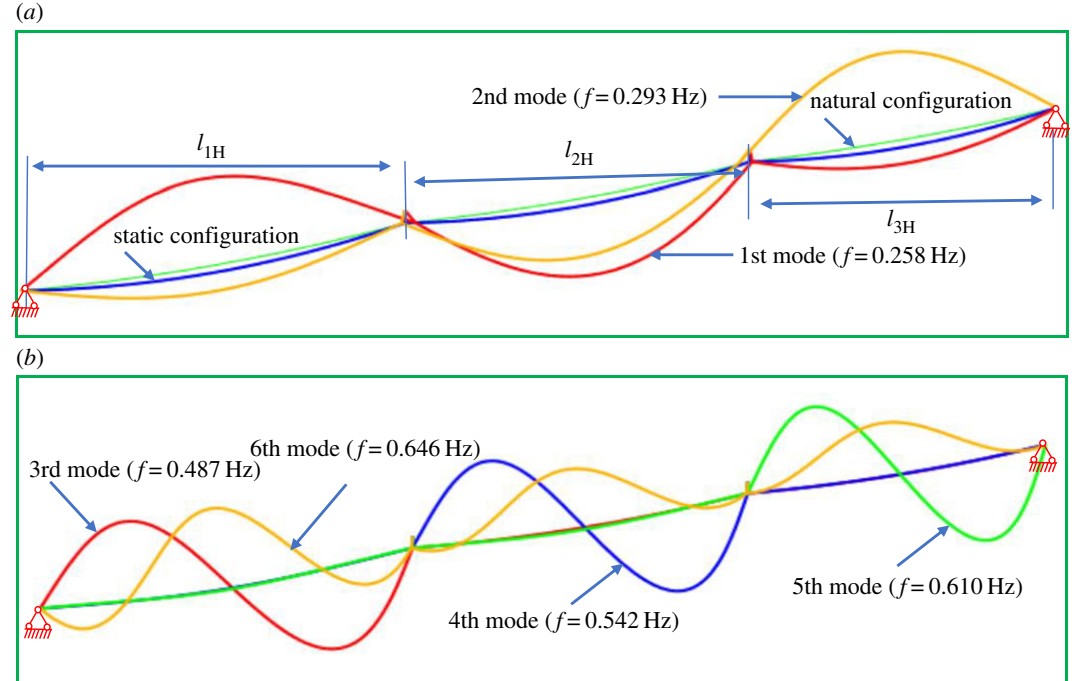

**Figure 15.** Different configurations of three-span transmission lines and corresponding modes determined by ABAQUS. (a) Different configurations and the first two modes. (b) Second, third, fourth, fifth and sixth modes.

The accuracy and advantages of the dynamic stiffness will be demonstrated by employing typical three-span transmission lines. The same material parameters as the previous example are used. The span length of three-span transmission lines is 200–180–150 m. The inclination angles of the first, second and third spans are the same as 20°. The length of suspension insulator string is 6.47 m. The horizontal tension of the first span in the static configuration under self-weight is usually known and is set to 14.25 kN($\lambda_1 = 3\pi$). The first six modes and natural frequencies calculated by FEM are shown in figure 15, and the dimensionless frequencies $\bar{\omega}_1/\pi$ are 1.055, 1.198, 1.992, 2.217, 2.495 and 2.642, respectively.

Substituting the parameters of the previous example into equation (3.14), the variation of the dynamic stiffness, $K_d$, with respect to the excited frequency is obtained. The dynamic stiffness is normalized with respect to the corresponding static stiffness value to calculate the ordinates, as shown in figure 16. The increment of the horizontal coordinate in the dynamic stiffness curve is 0.0005. The resonant peaks of the curves of the dynamic stiffness correspond to the dimensionless natural frequency of three-span transmission lines supported at both ends. The horizontal coordinates of the first four peaks, which are respectively 1.059, 1.207, 2.000, 2.222, 2.504 and 2.646, are very close to the dimensionless natural frequencies obtained by ABAQUS. The results show that the dynamic stiffness is more accurate by comparison with identifying natural frequency.

In order to study the effects of damping on the modal shape and natural frequency, the model of three-span transmission lines, in which the span lengths are 200–180–160 m and the inclination angles are 10°–10°–10°, have been established. The geometrical parameter $\lambda_1/\pi$ is set to 3. Damping of dynamic stiffness is specified by the dimensionless factor $\xi_1$, which is expressed in terms of the fundamental natural frequency of the associated first span lines. Figure 17a shows the real part of dynamic stiffness, and figure 17b shows the corresponding imaginary part of dynamic stiffness. Similar to the dynamic stiffness of two-span transmission lines, damping significantly affects the dynamic stiffness only at excited frequencies close to the natural frequencies of three-span transmission lines.

As presented in figure 18a, the solid lines are the first three symmetrical modes calculated by using ABAQUS, and the scatter plots are symmetrical modes calculated by using analytic solutions. The analytic solutions are similar to equation (2.30). The symmetrical modes calculated by using analytic solutions coincide well with the FEM results obtained by using ABAQUS. As shown in figure 18b, the first three antisymmetrical modes given by analytic solutions and ABAQUS are in good agreement.

Figure 19a shows real and imaginary component of the first symmetrical complex mode. It is shown that the damping parameter significantly affects the shape of the real component of the first symmetrical

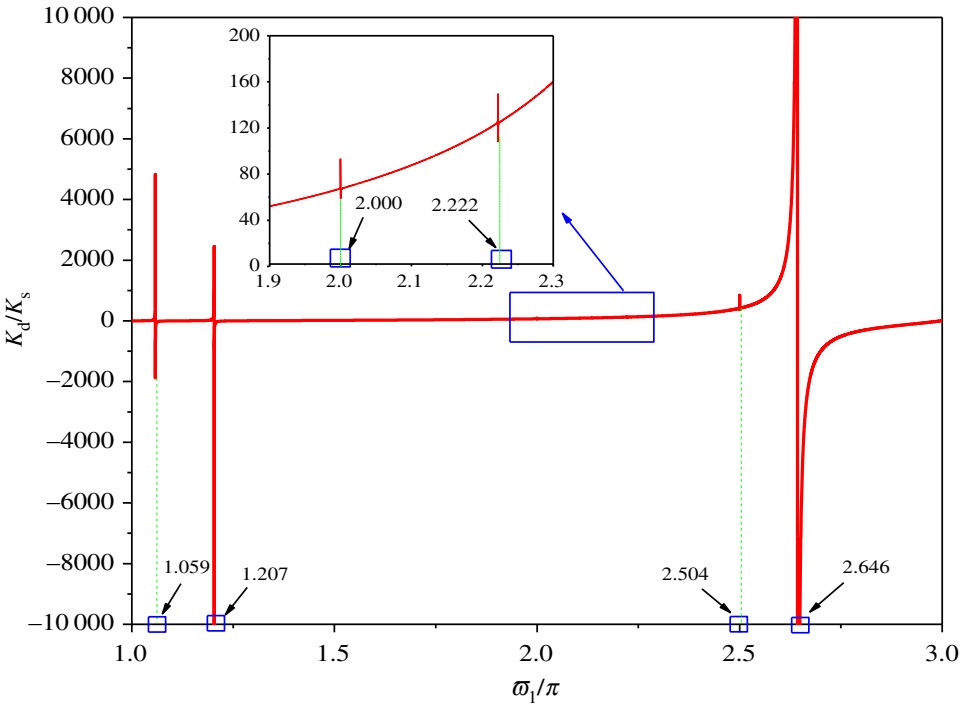

**Figure 16.** Variation of the dynamic stiffness of three-span transmission lines with excited frequency.

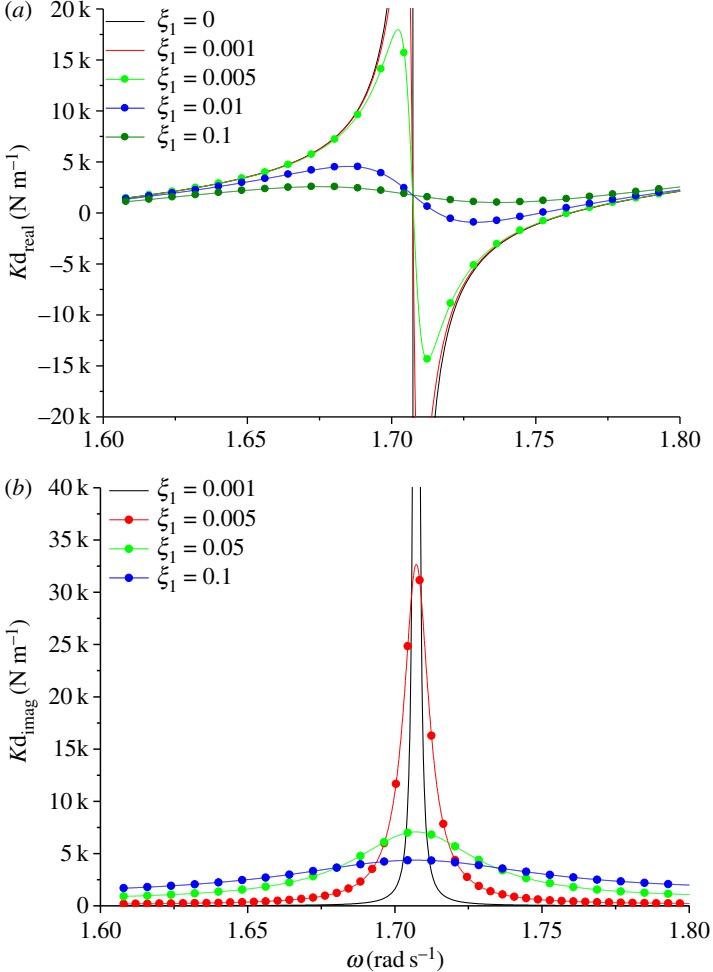

**Figure 17.** Dynamic stiffness of damped three-span transmission lines. (*a*) Real component. (*b*) Imaginary component.

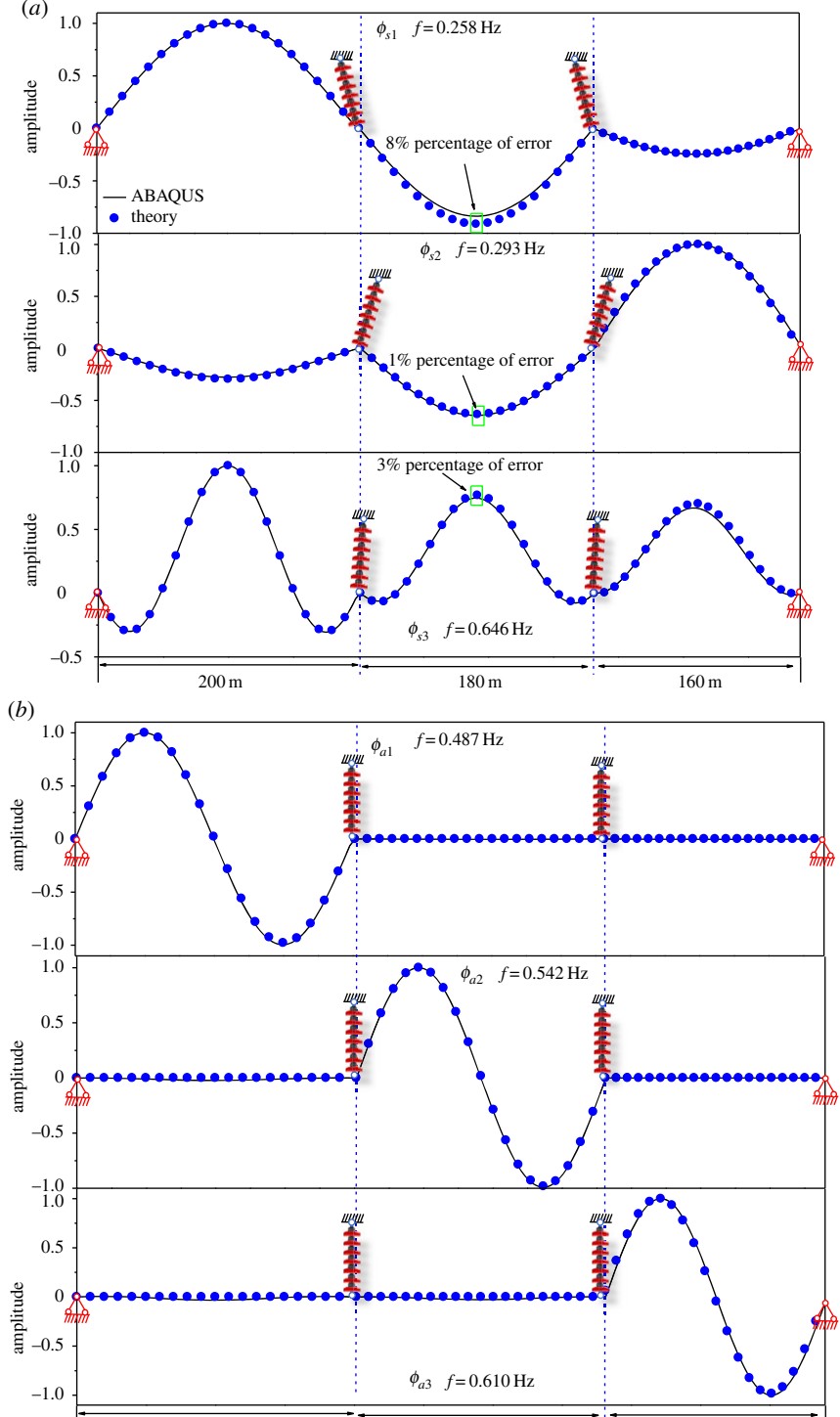

**Figure 18.** First six modal shapes of three-span with $\lambda_1/\pi = 3.0$. (a) Symmetrical modal shapes ($\phi_{s1}$, $\phi_{s2}$, $\phi_{s3}$). (b) Antisymmetrical modal shapes ($\phi_{a1}$, $\phi_{a2}$, $\phi_{a3}$).

complex mode when the value of the damping parameter is large. It is important to note that the shape of the real component of the first symmetrical complex mode is different from the shape of the imaginary component. Figure 19$b$ shows the real and imaginary components of the second symmetrical complex mode. Similarly, the damping parameter significantly affects the shape of the real component of the second symmetrical complex mode. Figure 13$c$ shows the real and imaginary components of the third symmetrical complex mode. It is shown that the damping parameter does not significantly affect the shape of the real component of the third symmetrical complex mode. The shape of the real

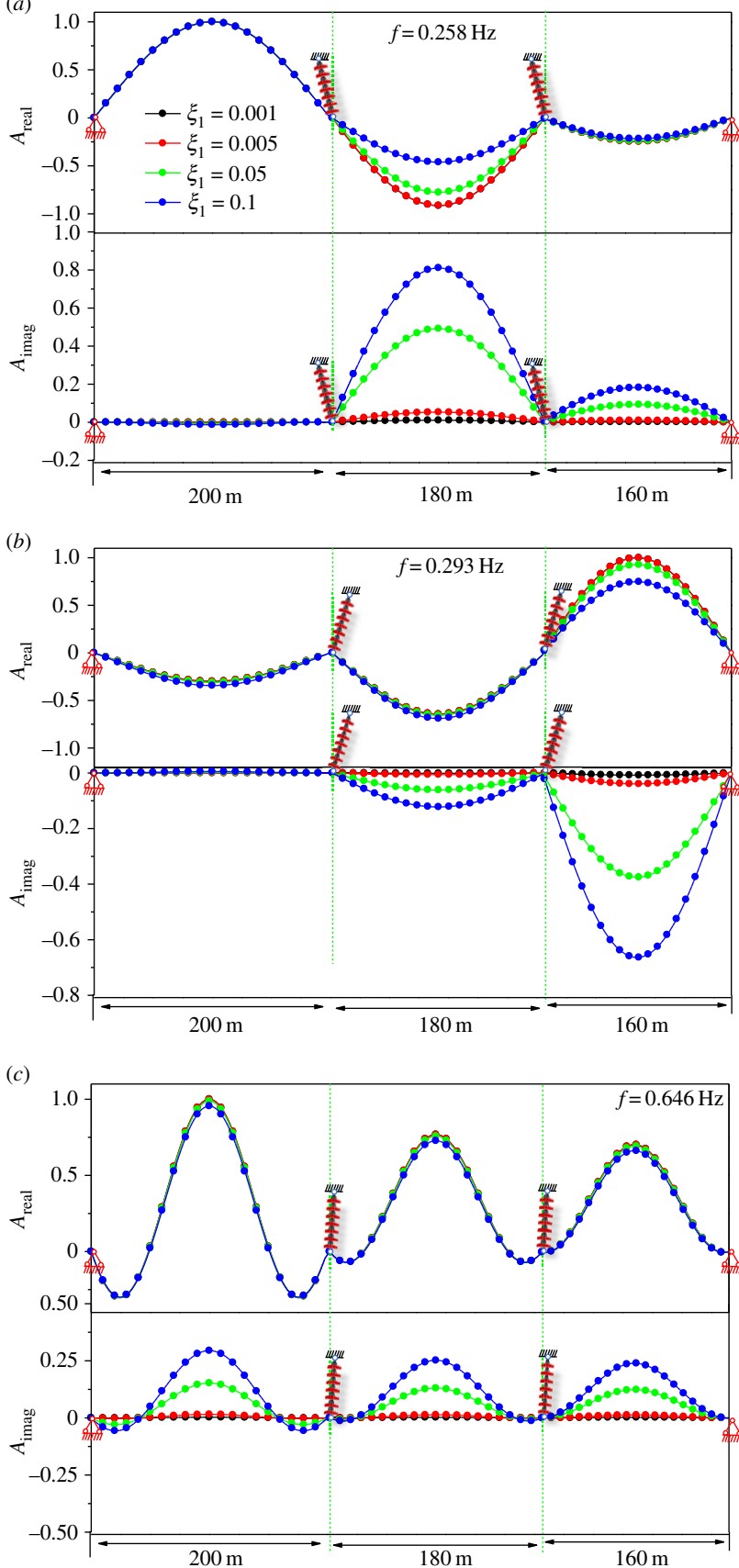

**Figure 19.** Modal shape of damped three-span transmission lines. (*a*) Modal shape of the first symmetrical mode. (*b*) Modal shape of the second symmetrical mode. (*c*) Modal shape of the third symmetrical mode.

component of the third symmetrical complex mode is close to that of the imaginary component. Considering the shape of the three-span antisymmetrical mode is similar to that of the two-span antisymmetrical mode, only the effect of the damping parameter on the symmetrical mode is studied.

## 4. Conclusion

The closed-form expressions for the equivalent dynamic stiffness have been developed to help designers assess the dynamic response of multi-span transmission lines more accurately and efficiently. The effects of adjacent spans and suspension insulator strings are incorporated in the equivalent dynamic stiffness to improve accuracy and to extend the range of application of the dynamic stiffness. The accuracy and advantages of the dynamic stiffness are demonstrated by employing some illustrative examples. The natural frequencies of multi-span transmission lines are identified on the dynamic stiffness curve with respect to excited harmonic frequency. For the multi-span transmission lines with inclination angle in the range between $-30°$ and $30°$, a comparison of the natural frequencies calculated by FEM and dynamic stiffness further demonstrates the rationality of dynamic stiffness. The parametrical study of dynamic stiffness of two-span transmission lines reveals that the dimensionless frequency of the first symmetric mode is small compared with other frequencies. The dynamic stiffness curve shows that the effect of the second symmetric mode on the dynamic tension is greater than that of the first symmetrical mode. When the motion of two-span transmission lines is dominated by the antisymmetrical in-plane mode, the change in horizontal dynamic tension is very small.

A detailed parametrical study shows that the length of insulator strings, geometrical parameter and inclination angles play important roles in the in-plane symmetrical modal shape and corresponding natural frequency. The model of multi-span transmission lines has several additional modes, due to interactions with the adjacent spans and insulators strings, which cannot be simulated by a single-span model. At certain values of the geometrical parameter, the natural frequency of a symmetric mode is close to that of an antisymmetric mode but they never coincide, while corresponding modes become hybrid modes, a mixture of symmetric and antisymmetric shapes. The lengths of two transition regions are very small, so the hybrid mode at the transition region can be obtained by employing a linear superposition of the symmetric and antisymmetric modes. The natural frequency identified by peak-to-peak of dynamic stiffness and corresponding modal shape can provide a good prediction of internal resonance condition and dynamic characteristics to guide the design of overhead transmission lines. The effect of damping on the dynamic stiffness and modal shape is studied. Some examples show that the damping significantly affects the dynamic stiffness and the modal shapes corresponding to low frequencies.

Data accessibility. Our data are deposited at Dryad Digital Repository: http://dx.doi.org/10.5061/dryad.3f31m01 [46].
Authors' contributions. X.H.L., Y.H. and M.Q.C. carried out the study of the dynamic stiffness, participated in the numerical simulation of the mode and natural frequency of transmission lines. X.H.L. and Y.H. analysed the data. X.H.L. interpreted the results and wrote the manuscript. All authors gave final approval for publication.
Competing interests. The authors declare no competing interests.
Funding. Financial support came from the National Natural Science Foundation of China and Basics and Cutting Edge Project of Chongqing Science and Technology Commission (grant no. 51507106, 51308570, and cstc2017jcyjAX0246)

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
