## [Reviewer comments · Royal Society Open Science]

Review History

RSOS-181354.R0 (Original submission)

Review form: Reviewer 1

Is the manuscript scientifically sound in its present form?

Yes

Are the interpretations and conclusions justified by the results?

Yes

Is the language acceptable?

Yes

Is it clear how to access all supporting data?

Yes

Do you have any ethical concerns with this paper?

No

Have you any concerns about statistical analyses in this paper?

No

Recommendation?

Accept with minor revision (please list in comments)

Comments to the Author(s)

In this paper, the dynamic stiffness of a two-span transmission line considering the interaction between the adjacent spans is studied, and the theoretical formulas for dynamic characteristics of the line are presented. The theory is verified through the comparison of theoretical solution and that by means of numerical method, and parameter study is also carried out. The method is extended to a multi-span line at the end. The topic is very interesting and the results are helpful for the understanding of free vibration behavior of multi-span transmission lines. The reviewer recommends the paper be published after some revision. Some comments are as the follows:

- (1) Page 2, line 41-42: the expression "since galloping is a motion of the cable very close to a free vibration" is unsuitable because galloping is actually a self-excited vibration.
- (2) What are the relative errors between the theoretical results (frequency and natural mode) and those determined by the numerical method in Fig.10?
- (3) In Fig. 2, the force on the right acting on the suspension insulator string should be H2 instead of H1?
- (4) In Eqs. (3a) and (3b), what are the definitions of the symbols l_{e1} and l_{e2} ?
- (5) Line246: T should be T1 ?
- (6) Line254: "1th" should be deleted.
- (7) In the finite element modeling, how many truss elements are used to simulate the suspension insulator string? Each porcelain insulator is simulated by one truss element?
- (8) In Fig.4, what is K_s ? is K_d the same as K_{dyn} in Eq.(24)?
- (9) Lines 528-531, the subscripts i and j should be consistent?
- (10) Line571: K_x should be K_{dyn} ?
- (11) In Section 3, the authors presented the dynamic stiffness of multi-span line. It is suggested that one or two examples be added to demonstrate the validation of the theoretical solution.
- (12) The sub-titles should be modified because the paper investigates not only the dynamic stiffness but also the free vibration characteristics of the transmission lines. It is suggested that Section 2 be divided into two sections, one discussing the dynamic stiffness and the other the free vibration characteristics.
- (13) The English expression should be improved.

Review form: Reviewer 2**Is the manuscript scientifically sound in its present form?**

Yes

Are the interpretations and conclusions justified by the results?

Yes

Is the language acceptable?

No

Is it clear how to access all supporting data?

No

Do you have any ethical concerns with this paper?

No

Have you any concerns about statistical analyses in this paper?

No

Recommendation?

Major revision is needed (please make suggestions in comments)

Comments to the Author(s)

The paper presents some analytical derivations to theoretically analyze the vibration response of transmission lines. The subject of the paper is of interest because of its significant applicative perspectives. The paper is well organized: the novel aspects of the study are clearly outlined in the introduction, and theoretical results are validated against numerical simulations in the subsequent sections; the influence of relevant parameters is also investigated. However, the last part of the paper discussing the extension of the method to multi-span conductors lacks of a validation example. In view of possible publication, the following comments have to be duly considered and the paper revised accordingly:

- A comparison of the analytical results with those obtained by a numerical simulation must be added for the multi-span case;
- A careful language editing is necessary: typos and misprints are disseminated throughout the manuscript;
- Symbols must be checked (see, for instance, lines 154, 173 and 203);
- The English language must be carefully revised, possibly with the aid of a native English speaker, because of the presence of several errors;
- Dimensions of all figures must be increased, and image resolution must be improved.

Review form: Reviewer 3

Is the manuscript scientifically sound in its present form?

No

Are the interpretations and conclusions justified by the results?

No

Is the language acceptable?

No

Is it clear how to access all supporting data?

Yes

Do you have any ethical concerns with this paper?

No

Have you any concerns about statistical analyses in this paper?

No

Recommendation?

Major revision is needed (please make suggestions in comments)

Comments to the Author(s)

The paper develops a dynamic stiffness model for multi-span transmission lines by including the insulator strings and inclination angle of the cables. Basically the methodology of a single span cable is extended to two-span and multi-span transmission systems. It is admitted that such an extension is helpful and far from being trivial, but the main criticism of the paper arises from the fact that the authors have not researched the effect of damping in any meaningful way. The paper cannot be accepted if damping is not given due recognition in an effective and rigorous manner, both in the theory and the results. The effect of damping on mode shapes must be elucidated. The discussion of results needs considerable sharpening. There are many presentational issues, particularly on the usage of the English language. Some, but not all, of them are as follows (the line number corresponds to the bold line number appearing in the second column from the left of the manuscript):

Line 8: characteristic should be characteristics

Lines 9 and 10: liens should be lines

Line 10: expression of should be expression for

Lines 17-18: The sentence is defective.

Lines 26-27: Reword the sentence.

Line 30: Fig.1 should be Figure 1 because it is the beginning of a sentence.

Line 42: Defective sentence.

Lines 44-45: Defective sentence.

Line 49: A should not be in capital in the middle of a sentence.

Line 63: characteristics, not characteristic.

Line 119: The caption of Fig. 2 should not be dynamic stiffness

Line 126: A reference for Eqs. 2a and 2b is needed.

Line 336: Figure caption misplaced.

Line 580: Replace ", this" by "which"

There are many other mistakes in the usage of the English language which must be rectified. At present the paper is not up to the standard of the Royal Society.

Decision letter (RSOS-181354.R0)

01-Nov-2018

Dear Dr Liu,

The editors assigned to your paper ("Free vibration analysis of transmission lines based on the dynamic stiffness method") have now received comments from reviewers. We would like you to revise your paper in accordance with the referee and Associate Editor suggestions which can be found below (not including confidential reports to the Editor). Please note this decision does not guarantee eventual acceptance.

Please submit a copy of your revised paper before 24-Nov-2018. Please note that the revision deadline will expire at 00.00am on this date. If we do not hear from you within this time then it will be assumed that the paper has been withdrawn. In exceptional circumstances, extensions may be possible if agreed with the Editorial Office in advance. We do not allow multiple rounds of revision so we urge you to make every effort to fully address all of the comments at this stage. If deemed necessary by the Editors, your manuscript will be sent back to one or more of the original reviewers for assessment. If the original reviewers are not available, we may invite new reviewers.

- Data accessibility

If you wish to submit your supporting data or code to Dryad (<http://datadryad.org/>), or modify your current submission to dryad, please use the following link:
<http://datadryad.org/submit?journalID=RSOS&manu=RSOS-181354>

- Competing interests

- Authors' contributions

- Acknowledgements

- Funding statement

Please note that Royal Society Open Science charge article processing charges for all new submissions that are accepted for publication. Charges will also apply to papers transferred to Royal Society Open Science from other Royal Society Publishing journals, as well as papers submitted as part of our collaboration with the Royal Society of Chemistry (<http://rsos.royalsocietypublishing.org/chemistry>). If your manuscript is newly submitted and subsequently accepted for publication, you will be asked to pay the article processing charge, unless you request a waiver and this is approved by Royal Society Publishing. You can find out more about the charges at <http://rsos.royalsocietypublishing.org/page/charges>. Should you have any queries, please contact openscience@royalsociety.org.

on behalf of Prof. R. Kerry Rowe (Subject Editor)
openscience@royalsociety.org

Associate Editor's comments:

We hope you find the comments of the referees to be of assistance in improving your manuscript, and bringing it to a standard acceptable for publication. Please ensure that you fully address the concerns of the referees in any revision. Please note that each referee indicates you should seek additional advice in improving the clarity and quality of the written English -- you may benefit from one of the services identified at <https://royalsociety.org/journals/authors/language-polishing/>. Please ensure that you provide evidence of having your work looked at by one of these (or an equivalent) service.

Thank you for your submission, and we look forward to receiving the revision, but please note that the journal does not generally allow multiple rounds of major revision, and if the referees remain unsatisfied by your modified manuscript, the manuscript may be rejected.

Comments to Author:

Reviewers' Comments to Author:

Reviewer: 1

Comments to the Author(s)

In this paper, the dynamic stiffness of a two-span transmission line considering the interaction between the adjacent spans is studied, and the theoretical formulas for dynamic characteristics of the line are presented. The theory is verified through the comparison of theoretical solution and that by means of numerical method, and parameter study is also carried out. The method is extended to a multi-span line at the end. The topic is very interesting and the results are helpful for the understanding of free vibration behavior of multi-span transmission lines. The reviewer recommends the paper be published after some revision. Some comments are as the follows:

- (1) Page 2, line 41-42: the expression "since galloping is a motion of the cable very close to a free vibration" is unsuitable because galloping is actually a self-excited vibration.
- (2) What are the relative errors between the theoretical results (frequency and natural mode) and those determined by the numerical method in Fig.10?
- (3) In Fig. 2, the force on the right acting on the suspension insulator string should be H2 instead of H1?
- (4) In Eqs. (3a) and (3b), what are the definitions of the symbols l_{e1} and l_{e2} ?
- (5) Line246: T should be T1 ?
- (6) Line254: "1th" should be deleted.
- (7) In the finite element modeling, how many truss elements are used to simulate the suspension insulator string? Each porcelain insulator is simulated by one truss element?
- (8) In Fig.4, what is K_s ? is K_d the same as K_{dyn} in Eq.(24)?
- (9) Lines 528-531, the subscripts i and j should be consistent?
- (10) Line571: K_x should be K_{dyn} ?
- (11) In Section 3, the authors presented the dynamic stiffness of multi-span line. It is suggested that one or two examples be added to demonstrate the validation of the theoretical solution.
- (12) The sub-titles should be modified because the paper investigates not only the dynamic stiffness but also the free vibration characteristics of the transmission lines. It is suggested that Section 2 be divided into two sections, one discussing the dynamic stiffness and the other the free vibration characteristics.
- (13) The English expression should be improved.

Reviewer: 2

Comments to the Author(s)

The paper presents some analytical derivations to theoretically analyze the vibration response of transmission lines. The subject of the paper is of interest because of its significant applicative perspectives. The paper is well organized: the novel aspects of the study are clearly outlined in the introduction, and theoretical results are validated against numerical simulations in the subsequent sections; the influence of relevant parameters is also investigated. However, the last part of the paper discussing the extension of the method to multi-span conductors lacks of a validation example. In view of possible publication, the following comments have to be duly considered and the paper revised accordingly:

- A comparison of the analytical results with those obtained by a numerical simulation must be added for the multi-span case;
- A careful language editing is necessary: typos and misprints are disseminated throughout the manuscript;
- Symbols must be checked (see, for instance, lines 154, 173 and 203);

- The English language must be carefully revised, possibly with the aid of a native English speaker, because of the presence of several errors;
- Dimensions of all figures must be increased, and image resolution must be improved.

Reviewer: 3

Comments to the Author(s)

The paper develops a dynamic stiffness model for multi-span transmission lines by including the insulator strings and inclination angle of the cables. Basically the methodology of a single span cable is extended to two-span and multi-span transmission systems. It is admitted that such an extension is helpful and far from being trivial, but the main criticism of the paper arises from the fact that the authors have not researched the effect of damping in any meaningful way. The paper cannot be accepted if damping is not given due recognition in an effective and rigorous manner, both in the theory and the results. The effect of damping on mode shapes must be elucidated. The discussion of results needs considerable sharpening. There are many presentational issues, particularly on the usage of the English language. Some, but not all, of them are as follows (the line number corresponds to the bold line number appearing in the second column from the left of the manuscript):

Line 8: characteristic should be characteristics

Lines 9 and 10: liens should be lines

Line 10: expression of should be expression for

Lines 17-18: The sentence is defective.

Lines 26-27: Reword the sentence.

Line 30: Fig.1 should be Figure 1 because it is the beginning of a sentence.

Line 42: Defective sentence.

Lines 44-45: Defective sentence.

Line 49: A should not be in capital in the middle of a sentence.

Line 63: characteristics, not characteristic.

Line 119: The caption of Fig. 2 should not be dynamic stiffness

Line 126: A reference for Eqs. 2a and 2b is needed.

Line 336: Figure caption misplaced.

Line 580: Replace ", this" by "which"

There are many other mistakes in the usage of the English language which must be rectified. At present the paper is not up to the standard of the Royal Society.

Author's Response to Decision Letter for (RSOS-181354.R0)

See Appendix A.

RSOS-181354.R1 (Revision)

Review form: Reviewer 2

Is the manuscript scientifically sound in its present form?

Yes

Are the interpretations and conclusions justified by the results?

Yes

Is the language acceptable?

Yes

Is it clear how to access all supporting data?

Not Applicable

Do you have any ethical concerns with this paper?

No

Have you any concerns about statistical analyses in this paper?

No

Recommendation?

Accept as is

Comments to the Author(s)

The Authors revised the paper following the reviewers' comments. The current version of the paper can be accepted for publication.

Review form: Reviewer 3

Is the manuscript scientifically sound in its present form?

Yes

Are the interpretations and conclusions justified by the results?

Yes

Is the language acceptable?

Yes

Is it clear how to access all supporting data?

Yes

Do you have any ethical concerns with this paper?

No

Have you any concerns about statistical analyses in this paper?

No

Recommendation?

Accept as is

Comments to the Author(s)

In the initial submission, damping was completely left out. As a consequence of my comment on the original manuscript, asking the authors to include damping, as it turned out, the revised version shows that the effects of damping could be quite pronounced. Lessons can be learned from this. The authors are requested to take cognizance of the fact that in an analysis of this type,

the effects of damping cannot be ignored. Also, it is noted with some dismay that research papers from some of the leading researchers in the dynamic stiffness method have not been quoted in the list of references. This is most unusual.

Decision letter (RSOS-181354.R1)

12-Feb-2019

Dear Dr Liu:

On behalf of the Editors, I am pleased to inform you that your Manuscript RSOS-181354.R1 entitled "Free vibration analysis of transmission lines based on the dynamic stiffness method" has been accepted for publication in Royal Society Open Science subject to minor revision in accordance with the referee suggestions. Please find the referees' comments at the end of this email.

The reviewers and Subject Editor have recommended publication, but also suggest some minor revisions to your manuscript. Therefore, I invite you to respond to the comments and revise your manuscript.

- Ethics statement

- Data accessibility

If you wish to submit your supporting data or code to Dryad (<http://datadryad.org/>), or modify your current submission to dryad, please use the following link:
<http://datadryad.org/submit?journalID=RSOS&manu=RSOS-181354.R1>

- Competing interests

- Authors' contributions

- Acknowledgements

- Funding statement

Because the schedule for publication is very tight, it is a condition of publication that you submit the revised version of your manuscript before 21-Feb-2019. Please note that the revision deadline will expire at 00.00am on this date. If you do not think you will be able to meet this date please let me know immediately.

- 1) A text file of the manuscript (tex, txt, rtf, docx or doc), references, tables (including captions) and figure captions. Do not upload a PDF as your "Main Document".
- 2) A separate electronic file of each figure (EPS or print-quality PDF preferred (either format should be produced directly from original creation package), or original software format)
- 3) Included a 100 word media summary of your paper when requested at submission. Please ensure you have entered correct contact details (email, institution and telephone) in your user account
- 4) Included the raw data to support the claims made in your paper. You can either include your data as electronic supplementary material or upload to a repository and include the relevant doi within your manuscript

5) All supplementary materials accompanying an accepted article will be treated as in their final form. Note that the Royal Society will neither edit nor typeset supplementary material and it will be hosted as provided. Please ensure that the supplementary material includes the paper details where possible (authors, article title, journal name).

on behalf of Prof R. Kerry Rowe (Subject Editor)
openscience@royalsociety.org

Associate Editor Comments to Author:

Thank you for the revision. As you'll see, the reviewers are more satisfied your manuscript is ready for acceptance. However, one of the reviewers notes your literature review does not include a number of key papers -- unfortunately, they have not provided more details, but it would be worth exploring the literature one more time to include additional context to support your work.

Reviewer comments to Author:

Reviewer: 3

Comments to the Author(s)

In the initial submission, damping was completely left out. As a consequence of my comment on the original manuscript, asking the authors to include damping, as it turned out, the revised version shows that the effects of damping could be quite pronounced. Lessons can be learned from this. The authors are requested to take cognizance of the fact that in an analysis of this type, the effects of damping cannot be ignored. Also, it is noted with some dismay that research papers from some of the leading researchers in the dynamic stiffness method have not been quoted in the list of references. This is most unusual.

Reviewer: 2

Comments to the Author(s)

The Authors revised the paper following the reviewers' comments. The current version of the paper can be accepted for publication.

Author's Response to Decision Letter for (RSOS-181354.R1)

See Appendix B.

Decision letter (RSOS-181354.R2)

18-Feb-2019

Dear Dr Liu,

I am pleased to inform you that your manuscript entitled "Free vibration analysis of transmission lines based on the dynamic stiffness method" is now accepted for publication in Royal Society Open Science.

on behalf of Prof R. Kerry Rowe (Subject Editor)
openscience@royalsociety.org

Appendix A

Dear Editors:

Thank you very much for your email forwarding the referees' reviews and valuable comments on the paper "Free vibration analysis of transmission lines based on the dynamic stiffness method". We have studied your comments carefully and have made correction which we hope to meet with your approval.

According to the Reviewer #1's suggestion, we made some revisions and would like to add following notes:

(1) Page 2, line 41-42: the expression "since galloping is a motion of the cable very close to a free vibration" is unsuitable because galloping is actually a self-excited vibration.

According to the reviewer's suggestion, the expression "since galloping is a motion of the cable very close to a free vibration" has been corrected to "since some galloping characteristics are similar to those of free vibrating suspended cable." in the revised manuscript.

(2) What are the relative errors between the theoretical results (frequency and natural mode) and those determined by the numerical method in Fig.10?

According to the reviewer's suggestion, the percentage errors between the modes calculated by theoretical method and FEM have been added in the revised manuscript, as shown in Fig.10. The relative errors between the theoretical natural frequency and those determined by the numerical method are presented in Tab.1-4.

(3) In Fig. 2, the force on the right acting on the suspension insulator string should be H2 instead of H1?

According to the reviewer's suggestion, the "H1" has been corrected to "H2" in the revised manuscript.

(4) In Eqs. (3a) and (3b), what are the definitions of the symbols l_{e1} and l_{e2} ?

According to the reviewer's suggestion, the definitions of the symbols l_{e1} and l_{e2} have been added in the revised manuscript. The added content is as follows: l_{e1} and l_{e2} are the effective cable lengths of the 1st and 2nd spans, respectively.

(5) Line246: T should be T_1 ?

Thanks for the reviewer's precious proposal, T has been corrected to T_1 in the revised manuscript.

(6) Line254: "1th" should be deleted.

According to the reviewer's suggestion, "1th" has been deleted in the revised manuscript.

(7) In the finite element modeling, how many truss elements are used to simulate the suspension insulator string? Each porcelain insulator is simulated by one truss element?

Twenty eight truss elements are used to simulate the suspension insulator string. Each porcelain insulator is simulated by one truss element. The corresponding content is added in revised manuscript.

(8) In Fig.4, what is K_s ? is K_d the same as K_{dyn} in Eq.(24)?

Thanks for the reviewer's precious proposal, K_s is the static stiffness. The static stiffness may be obtained from Eq.(24) by letting $\omega=0$. The corresponding content is added in the revised manuscript. K_d has the same meaning as K_{dyn} . K_{dyn} is replaced by K_d for easy reading in the revised manuscript.

(9) Lines 528-531, the subscripts i and j should be consistent?

Thanks for the reviewer's precious proposal, the subscripts i and j have been consistent, the subscript " j " has been replaced to " i " in the revised manuscript.

(10) Line571: K_x should be K_{dyn} ?

K_x has the same meaning as K_{dyn} . K_x is replaced by K_d in the revised manuscript.

(11) In Section 3, the authors presented the dynamic stiffness of multi-span line. It is suggested that one or two examples be added to demonstrate the validation of the theoretical solution.

According to the reviewer's suggestion, one example has been added to demonstrate the validation of the theoretical solution in section 3 in the revised manuscript.

(12) The sub-titles should be modified because the paper investigates not only the dynamic stiffness but also the free vibration characteristics of the transmission lines. It is suggested that Section 2 be divided into two sections, one discussing the dynamic stiffness and the other the free vibration characteristics.

According to the reviewer's suggestion, the sub-titles has been modified, the sub-title has been changed to "Dynamic stiffness and modal function of two-span transmission lines".

(13) The English expression should be improved.

Thanks for the reviewer's precious proposals. After carefully proofread and checked the manuscript, some inappropriate expressions have been corrected in revised manuscript in red color.

According to the Reviewer #2's suggestion, we made some revisions and would like to add following notes:

(1)A comparison of the analytical results with those obtained by a numerical simulation must be

added for the multi-span case;

According to the reviewer's suggestion, one example has been added to demonstrate the validation of the multi-span theoretical solution in section 3 in revised manuscript.

(2) A careful language editing is necessary: typos and misprints are disseminated throughout the manuscript;

According to the reviewer's suggestion, typos and misprints have been carefully revised in revised manuscript.

(3) Symbols must be checked (see, for instance, lines 154, 173 and 203);

According to the reviewer's suggestion, we have thoroughly checked the symbols. The symbols that are inconsistent with the description of the paper are corrected in the revised manuscript.

(4) The English language must be carefully revised, possibly with the aid of a native English speaker, because of the presence of several errors;

According to the reviewer's suggestion, the manuscript has been carefully proofread and checked, some inappropriate expressions have been corrected in revised manuscript.

(5) Dimensions of all figures must be increased, and image resolution must be improved.

According to the reviewer's suggestion, all figures and images have been improved and modified.

According to the Reviewer #3's suggestion, we made some revisions and would like to add following notes:

(1) It is admitted that such an extension is helpful and far from being trivial, but the main criticism of the paper arises from the fact that the authors have not researched the effect of damping in any meaningful way. The paper cannot be accepted if damping is not given due recognition in an effective and rigorous manner, both in the theory and the results. The effect of damping on mode shapes must be elucidated. The discussion of results needs considerable sharpening.

According to the reviewer's suggestion, the effect of damping on the modal shape and the dynamic stiffness have been elucidated through two-span and three-span transmission lines, respectively. Corresponding content have been added in section 2 and 3 in the revised manuscript. The results showed that the damping parameter affected significantly the shape of the real component of the symmetrical complex mode when the value of damping parameter was large.

(2) Line 8: characteristic should be characteristics

According to the reviewer's suggestion, the "characteristic" has been corrected to "characteristics"

in revised manuscript.

(3) Lines 9 and 10: liens should be lines

According to the reviewer's suggestion, the "liens" has been corrected to "lines" in revised manuscript.

(4) Line 10: expression of should be expression for

According to the reviewer's suggestion, the "expression of" has been corrected to "expression for" in revised manuscript.

(5) Lines 17-18: The sentence is defective.

According to the reviewer's suggestion, the "a simplified modal function corresponding natural frequency is derived" has been corrected to "explicit formulae are derived for the modal shapes corresponding natural frequencies."

(6) Lines 26-27: Reword the sentence.

According to reviewer's suggestion, the sentence has been corrected to "The dynamic stiffness can be applied to evaluate the dynamic response of cables and of cable-supported structures in any cable-structure system."

(7) Line 30: Fig.1 should be Figure 1 because it is the beginning of a sentence.

According to reviewer's suggestion, the "Fig.1" has been corrected to "Figure 1" in revised manuscript.

(8) Line 42: Defective sentence.

According to reviewer's suggestion, the defective sentence has been corrected to "For instance, the frequency and mode shape of a galloping conductor are similar to those of free vibrating suspended cable, and internal resonance phenomena of suspended cable induced by galloping can be observed from free vibrating suspended cable."

(9) Lines 44-45: Defective sentence.

According to reviewer's suggestion, the defective sentence has been corrected to "Irvine presented a linear theory for free vibration of a horizontal single-span cable fixed at both ends and discussed the effect of the geometrical-elastic parameter λ on the modal shape and natural frequency."

(10) Line 49: A should not be in capital in the middle of a sentence.

According to reviewer's suggestion, the "A" has been corrected to "a".

(11) Line 63: characteristics, not characteristic.

According to reviewer's suggestion, the "characteristic" has been corrected to "characteristics".

(12) Line 119: The caption of Fig. 2 should not be dynamic stiffness

According to reviewer's suggestion, the caption of Fig.2 has been corrected to "A schematic of two-span transmission line with insulator string under external excitations.

(13) Line 126: A reference for Eqs. 2a and 2b is needed.

According to reviewer's suggestion, the reference for Eqs.2a and 2b has been added in revised manuscript.

(14) Line 336: Figure caption misplaced.

According to reviewer's suggestion, the Fig.5 caption has been placed in the corresponding position.

(15) Line 580: Replace ", this" by "which" There are many other mistakes in the usage of the English language which must be rectified.

According to reviewer's suggestion, the " this " has been corrected to the "which" in line 580. Other similar issues have been carefully reviewed and revised.

Appendix B

Dear Editors:

Thank you very much for your email forwarding the referees' reviews and valuable comments on the paper "Free vibration analysis of transmission lines based on the dynamic stiffness method". We have studied your comments carefully and have made correction which we hope to meet with your approval.

According to the associate editor's suggestion, we made some revisions and would like to add following notes:

(1) In response to the associate editor's comment "your literature review does not include a number of key papers -- unfortunately, they have not provided more details, but it would be worth exploring the literature one more time to include additional context to support your work."

Thanks for the associate editor's precious proposal, some literatures have been added in the literature review of the revised manuscript to support my research work. Modified contents had been corrected in revised manuscript in red color.

According to the Reviewer #3's suggestion, we made some revisions and would like to add following notes:

1. In response to the reviewer #3's comment "Also, it is noted with some dismay that research papers from some of the leading researchers in the dynamic stiffness method have not been quoted in the list of references. This is most unusual."

Thanks for the reviewer's precious proposal, the research papers from some of the leading researchers in the dynamic stiffness method were added to the references in the revised manuscript. Modified contents had been corrected in revised manuscript in red color.